# A comparison of sea salt emission parameterizations in Northwestern Europe using a chemistry transport model setup

Neumann Daniel[1], Matthias Volker[1], Bieser Johannes[1,2], Aulinger Armin[1], and Quante Markus[1]

[1]Helmholtz-Zentrum Geesthacht, Institute of Coastal Research, Max-Planck-Straße 1, 21502 Geesthacht, Germany
[2]Deutsches Zentrum für Luft- und Raumfahrt, Institute of Atmospheric Physics, Münchener Straße 20, 82234 Weßling, Germany

*Correspondence to:* daniel.neumann@hzg.de

**Abstract.**

Atmospheric sea salt particles affect chemical and physical processes in the atmosphere. These particles provide surface area for condensation and reaction of nitrogen, sulfur, and organic species and are a vehicle for the transport of these species. Additionally, HCl is released from sea salt. Hence, sea salt has a relevant impact on air quality, particularly in coastal regions with high anthropogenic emissions, such as the North Sea region. Therefore, the integration of sea salt emissions in modeling studies in these regions is necessary. However, it was found that sea salt concentrations are not represented with the necessary accuracy in some situations.

In this study, three sea salt emission parameterizations depending on different combinations of wind speed, salinity, sea surface temperature, and wave data were implemented and compared: GO03 (Gong, 2003), SP13 (Spada et al., 2013), and OV14 (Ovadnevaite et al., 2014). The aim was to identify the parameterization that most accurately predicts the sea salt mass concentrations at different distances to the source regions. For this purpose, modeled particle sodium concentrations, sodium wet deposition, and aerosol optical depth were evaluated against measurements of these parameters. Each two months in winter and summer 2008 were considered for this purpose. The shortness of these periods limits generalizability of the conclusions on other years.

While the GO03 emissions yielded overestimations in the $PM_{10}$ concentrations at coastal stations and underestimations of those at inland stations, OV14 emissions conversely led to underestimations at coastal stations and overestimations at inland stations. Because of the differently shaped particle size distributions of the GO03 and OV14 emission cases, the deposition velocity of the coarse particles differed between both cases which yielded this distinct behavior at inland and coastal stations. The $PM_{10}$ concentrations produced by the SP13 emissions generally overestimated the measured concentrations. The sodium wet deposition was generally underestimated by the model simulations but the SP13 cases yielded the least underestimations. Because the model tends to underestimate wet deposition, this result needs to be considered critically. Measurements of the aerosol optical depth (AOD) were underestimated by all model cases in the summer and partly in winter. None of the model cases clearly improved the modeled AODs. Overall, GO03 and OV14 produced the most accurate results, but both parameterizations revealed weaknesses in some situations.

# 1 Introduction

Sea salt particles affect atmospheric chemistry (Seinfeld and Pandis, 2006) and cloud formation. These particles are emitted as water droplets from the sea surface as a result of strong winds, the breaking of waves and the bursting of air bubbles. The parameterization of sea salt emissions has a long history (e.g., Blanchard and Woodcock, 1980; Fairall et al., 1983; Monahan and Muircheartaigh, 1980). Such parameterizations are necessary in chemistry transport models (CTMs) and climate models because sea salt particles impact atmospheric processes. Extensive reviews of sea salt emissions and emission parameterizations have been published in recent years (Lewis and Schwartz, 2004; de Leeuw et al., 2011; O'Dowd and de Leeuw, 2007; Spada et al., 2013).

Sea salt particles generated by the bursting of bubbles are the most relevant for atmospheric chemistry because they are smaller than sea salt particles produced by other processes, and thus, they have the longest atmospheric lifetime: air is entrained in the sea water by the breaking of waves, which is primarily wind driven, and forms air bubbles, which then rise to the surface where they burst (Monahan et al., 1986). Organic surfactants at the surface, the sea surface temperature (SST) and the sea surface salinity (SAL) affect these processes (Mårtensson et al., 2003; Salter et al., 2015; Blanchard, 1964; Donaldson et al., 2006). A large number of parameterizations relating sea salt emissions to wind speed and other parameters have been published in recent decades. Several were derived from a wind-speed-based parameterization published by Monahan and Muircheartaigh (1980) and Monahan et al. (1986). Nevertheless, atmospheric sea salt concentrations are still not satisfactorily reproduced by CTMs (e.g., Chen et al., 2016; Neumann et al., 2016; Gantt et al., 2015; Im, 2013; Spada et al., 2013; Tsyro et al., 2011), and improving these predictions remains an objective of ongoing research (Ovadnevaite et al., 2014; Gantt et al., 2015; Petelski et al., 2014; Salter et al., 2015; Long et al., 2011).

The North and Baltic Sea regions are areas of high anthropogenic activity giving rise to the emission of various air pollutants such as nitrogen oxides ($NO_x$), sulfur dioxide ($SO_2$), ammonia ($NH_3$) and primary particulate matter, which lead to the formation of nitric acid ($HNO_3$), sulfuric acid ($H_2SO_4$) and secondary particulate matter. The condensation of $HNO_3$, $H_2SO_4$ and $NH_3$ onto sea salt affects their atmospheric lifetimes and deposition patterns. The latter are important for quantifying the input of pollutants and nutrients into water bodies, e.g., for studying eutrophication. Additionally, the condensation of $H_2SO_4$ and $HNO_3$ leads to a release of HCl into the atmosphere (chlorine displacement/depletion). Hence, sea salt plays an important role in affecting the composition of particulate matter as well as the deposition and heterogeneous chemistry of relevant pollutants in this air pollution regime (Chen et al., 2016; Neumann et al., 2016; Crisp et al., 2014; Im, 2013; Kelly et al., 2010; Athanasopoulou et al., 2008). Therefore, when modeling air pollution in Northwestern Europe, sea salt emissions must be adequately parameterized.

The purpose of this study is to improve modeled atmospheric sea salt concentrations in Northwestern Europe by evaluating various open-ocean sea salt emission parameterizations and suggesting improvements for sea salt emissions. This is performed by comparing three different sea salt emission parameterizations (Gong, 2003; Spada et al., 2013; Ovadnevaite et al., 2014) with each other as well as with sodium concentrations and wet deposition measurements from stations within the network of the European Measurement and Evaluation Programme (EMEP) and with aerosol optical depth measurements by stations of

the Aerosol Robotic Network (AERONET). Gong (2003), which describes sea salt emissions by bubble bursting, is a widely used parameterization that depends only on the wind speed. Spada et al. (2013) compared several parameterizations from which MA03/MO86/SM93 is used here. This parameterization depends on wind speed and SST. In addition to Gong (2003), Spada et al. (2013) described the emission of spume droplets for high wind speeds. The parameterizations of Gong (2003) and Spada et al. (2013) were extended to depend on salinity (Neumann et al., 2016). This approach considerably improved the sodium concentrations predicted in the Baltic Sea region. Ovadnevaite et al. (2014) is a quite new parameterization that depends on wind speed, SST, SAL, and wave data and that should cover all sea salt production processes. This parameterization has not been used in a CTM setup in the study region.

There have been a few recent studies on sea salt in the Northwestern European region. Manders et al. (2010) evaluated sea salt measurements from various EMEP stations. The sea salt emissions is the Baltic Sea region were reduced by $90\,\%$ in that study. This result is notable because most studies do not consider variations in the salinity at all. Other studies addressed data from Mace Head (Cavalli et al., 2004; Ovadnevaite et al., 2014) and derived a new sea salt emission parameterization (Ovadnevaite et al., 2014). Tsyro et al. (2011) compared five open-ocean sea salt emission parameterizations, which depended on the wind speed only, in Europe. Chen et al. (2016) in detail evaluated particulate sodium concentrations predicted by a coupled meteorology chemistry transport model against measurements of an 11-day measurement campaign. Finally, Im (2013) performed and evaluated CTM simulations in the southeastern European region with a focus on particulate sodium and nitrate. Both latter publications did consider only one sea salt emission parameterization. In this study comparing three sea salt emission parameterizations, the impact of SAL on generation of sea salt particles and the contribution of surf zone emissions are considered. Additionally, using the parameterization of Ovadnevaite et al. (2014), an explicitly wave-dependent function is considered in this study.

## 2  Materials and Methods

### 2.1  Chemistry Transport Model

The simulations were performed using the Community Multiscale Air Quality (CMAQ) modeling system, which is developed and maintained by the U.S. EPA. Version 5.0.1 was used in this study. The study region was enclosed by a grid with dimensions of $24 \times 24\ \mathrm{km}^2$, which was one-way nested in a coarse grid with dimensions of $72 \times 72\ \mathrm{km}^2$ (Fig. 1). The outer boundary conditions were taken from TM5 model runs (Huijnen et al., 2010). The cb05tucl mechanism (Yarwood et al., 2005; Whitten et al., 2010; Tanaka et al., 2003; Sarwar et al., 2007) was used to represent the gas-phase chemistry, and the AERO05 mechanism (Nenes et al., 1998, 1999) was used to represent the particle-phase chemistry. CMAQ also includes in-cloud chemistry. The dry deposition parameterization for particulate matter is an updated version of Binkowski and Shankar (1995), which is based on Slinn and Slinn (1980) and Pleim et al. (1984). The parameterization considers gravitational settling, aerodynamic resistance above the canopy, and surface resistance. The three modes and the three moments are deposed individually. The particle representation by modes and moments is described further below.

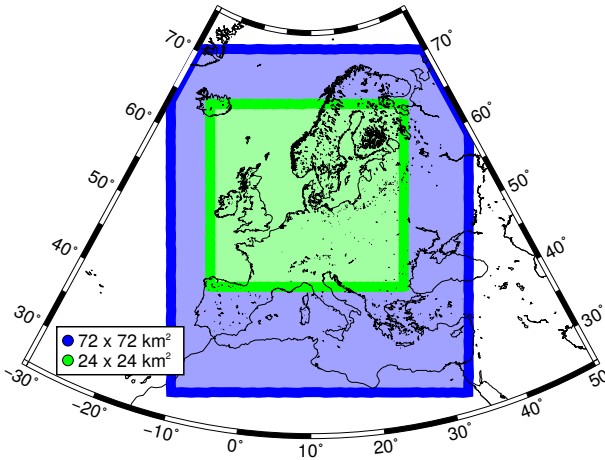

**Figure 1.** Study region and size of the model grids. The coarse grid (blue) includes Europe and parts of northern Africa. The first nested grid (green) includes Northwestern Europe, including the North and Baltic Seas.

Dust was not included in either the boundary conditions or the emissions. The dust concentrations in Northwestern Europe are low compared to sea salt and anthropogenic particle concentrations (Cuevas et al., 2015). Moreover, in episodes with high dust loading, Sahara dust is commonly transported in higher atmospheric layers across Northwestern Europe according to MACC II data (obtained from http://macc.copernicus-atmosphere.eu).

The aerosol phase is represented by three log-normally distributed modes: the Aitken, accumulation and coarse modes. Each size mode is represented by three moments (3-moment scheme): the total particle number (0th moment), the total particle surface area ($2\pi$ of the 2nd moment), and the total particle mass ($\frac{4}{3}\pi \times \rho_{ss}$ of the 3rd moment; $\rho_{ss}$ = sea salt dry density). The total mass is split into speciated mass fractions, but the total number and surface area emissions are not. The standard deviation and geometric mean diameter (GMD) of each size mode are not fixed but rather are calculated from the moments

when necessary. Binkowski and Roselle (2003) and the CMAQ Wiki (http://www.airqualitymodeling.org/cmaqwiki) describe the CMAQ aerosol mechanism in greater detail.

## 2.2   Sea Salt Emissions

In this section, sea salt emissions are described from three perspectives: (1) the physical processes related to sea salt emissions, (2) the sea salt emission parameterizations compared in this study, and (3) the technical implementation of the sea salt emission

parameterizations for CMAQ.

### 2.2.1 Physical Processes

Water droplets are emitted from the sea surface by the bursting of bubbles (film and jet droplets), by the breaking of waves (splash droplets) and by high wind speeds (spume droplets). The droplet water evaporates until the droplet water content is in equilibrium with the ambient relative humidity. This droplet is denoted as wet sea salt particles.

When air is mixed into sea water by processes such as the breaking of waves, the air forms bubbles, which then rise to the sea surface where they burst. Small water droplets are ejected from the breaking hull of a bubble (film droplets). Because of the abrupt change in pressure within the bursting bubble, water is also sucked from below the bubble into the air (jet droplets). The bursting of bubbles is the most relevant process for the production of sea salt particles. An increase in wind speed increases wave generation, wave breaking, and, consequently, bubble-bursting-generated sea salt emissions. Sea salt particles from spume and splash droplets are very large and commonly fall back into the ocean within a short time after their emission. They are only relevant at high wind speeds (Lewis and Schwartz, 2004). The SST affects the formation and bursting of air bubbles (Mårtensson et al., 2003; Callaghan et al., 2014; Grythe et al., 2014), thereby altering the size distribution of the sea salt particles thus produced. Changing the SAL also alters the particle size – a lower salinity leads to smaller particles (Mårtensson et al., 2003). Moreover, organic species are relevant to sea salt emissions, but their actual impact has not yet been well quantified.

In the surf zone, which is the region along a coast line where waves break, sea salt emissions are enhanced because of the higher number of breaking waves in this relatively small region. Addressing surf zone emissions is quite difficult because they depend on the direction of the waves, the direction of the wind, and local coastal features such as steep cliff coasts and flat beaches.

### 2.2.2 Sea Salt Emission Parameterizations

The existing sea salt emission parameterizations and their historical development have been extensively described and compared in Lewis and Schwartz (2004), O'Dowd and de Leeuw (2007), de Leeuw et al. (2011), Tsyro et al. (2011), and Spada et al. (2013).

Three parameterizations, developed by Gong (2003), Spada et al. (2013), and Ovadnevaite et al. (2014), and a reference case without any sea salt emissions are compared in this study. These cases are denoted as GO03, SP13, OV14, and zero, respectively. GO03 is the standard parameterization in CMAQ (Kelly et al., 2010). SP13 consists of three existing parameterizations proposed by Mårtensson et al. (2003) (MA03), Monahan et al. (1986) (MO86), and Smith et al. (1993) (SM93). Table 1 presents an overview of these parameterizations. Relevant aspects thereof are described below. The formulas are provided in the appendix (Eqs. (B1) to (B7)). A more detailed description of the formulas and of their implementation are provided in the supplement to this paper (Sect. S1).

All three parameterizations describe the size distribution of sea salt particle emissions in terms of number. For their implementation in CMAQ, log-normal distributions are preferred. GO03 is represented by two log-normal distributions in CMAQ and it describes the bubble-generated production of sea salt particles. SP13 consists of a combination of different types of

**Table 1.** Overview of sea salt emission parameterizations GO03, SP13, and OV14.

| Parameter-ization | Functional Relation | Wind Dependence | Surf Zone | Parameters | Range of Validity | Reference |
|---|---|---|---|---|---|---|
| GO03 | two log-normal distributions | MO80 | KE10 | $u_{10}$, SAL[a] | $0.07\,\mu m \leq D_{dry} \leq 20\,\mu m$ | Gong (2003) |
| SP13 | mixed | mixed | mixed | $u_{10}$, SST, SAL[a] | $0.02\,\mu m \leq D_{dry} \leq 30\,\mu m$ | Spada et al. (2013) |
| MA03 | three polynomials | MO80 | KE10 | $u_{10}$, SST, SAL[a] | $0.02\,\mu m \leq D_{dry} \leq 2.8\,\mu m$ | Mårtensson et al. (2003) |
| MO86 | special function | MO80 | KE10 | $u_{10}$, SAL[a] | $2.8\,\mu m \leq D_{dry} \leq 8\,\mu m$ [b] | Monahan et al. (1986) |
| SM93 | two log-normal distributions | own: wind | no | $u_{10}$, SAL[a] | $2.8\,\mu m \leq D_{dry} \leq 30\,\mu m$ and $u_{10} \geq 9\,m\,s^{-1}$ | Smith et al. (1993) |
| OV14 | five log-normal distributions | own: wind and waves | no | $u_{10}$, $H_S$, $u_*$ SAL, SST | $0.015\,\mu m \leq D_{dry} \leq 6\,\mu m$ $Re_{Hw} \geq 10^5$ [c] | Ovadnevaite et al. (2014) |

[a] Originally, the parameterization does not depend on the salinity (SAL). The SAL dependence was added in this study (see Neumann et al., 2016). [b] MO86 is valid on the size range $2.8\,\mu m \leq D_{dry} \leq 8\,\mu m$ if it is not used in this context. [c] The fifth mode is only valid for $Re_{Hw} \geq 2 \times 10^5$.

Abbreviations: MO80 refers to Monahan and Muircheartaigh (1980), KE10 refers to Kelly et al. (2010), $u_{10} = 10$ m wind speed, SAL = sea surface salinity, SST = sea surface temperature, $H_S$ = significant wave height, $u_*$ = friction velocity at sea surface, $D_{dry}$ = dry sea salt particle diameter.

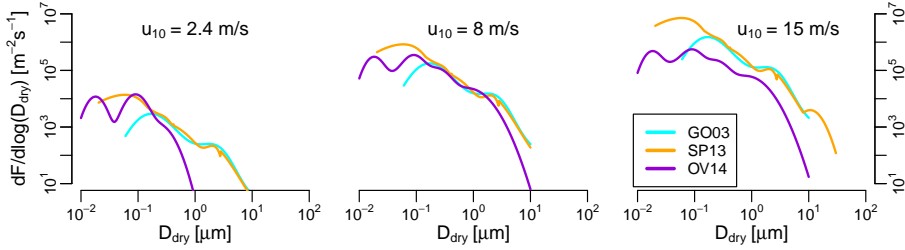

**Figure 2.** Comparison of the source functions and their wind speed dependence. The largest size mode of OV14 is deactivated for $Re_{Hw} \leq 2 \times 10^5$, and all modes are deactivated for $Re_{Hw} \leq 10^5$ (see Eq. (B7) for the definition of $Re_{Hw}$). In SP13, a spume-droplet-generated mode represented by SM93 is activated for $u_{10} \geq 9\,m\,s^{-1}$. The parameters used were as follows: SST = 283 K, SAL = 35‰, $C_D = 2.15 \times 10^{-3}$, $H_S = 1.23$ m, and $\nu_w = 1.34 \times 10^{-6}\,m^2\,s^{-1}$

functions and cannot be simply represented using log-normal distributions. It describes the production of sea salt particles generated by bursting bubbles (MA03 and MO86) and spume droplets (SM93). Spume droplet production is activated at wind speeds above $9\,m\,s^{-1}$ (Monahan et al., 1986). MA03 is based on laboratory studies. Finally, OV14 is a linear combination of five log-normal distributions. It describes bubble-bursting- and spume-droplet-generated sea salt emissions and is based on

5   measurements recorded at Mace Head, Ireland.

The wind speed dependence of GO03 and SP13 (MA03 and MO86) is described by the whitecap coverage parameterization proposed by Monahan and Muircheartaigh (1980). The parameterization relates the 10 m wind speed, $u_{10}\,[m\,s^{-1}]$, to the fraction of the sea surface covered by whitecaps, denoted by the whitecap coverage W. Bubble bursting and consequently sea

salt production depend linearly on the whitecap coverage. $W$ (Eq. (1)) scales the distribution functions but does not alter their shape. OV14 employs another wind speed dependence. Each of the five modes is scaled by an individual power-law function that depends on a Reynolds number, $\mathrm{Re_{Hw}}$, which is calculated from the friction velocity at the sea surface, $u_*$ $[\mathrm{m\ s^{-1}}]$; the significant wave height, $H_S$ $[\mathrm{m}]$; and the sea water kinetic viscosity, $\nu_w$ $[\mathrm{m^2\ s^{-1}}]$. The parameter $u_*$ is calculated from $u_{10}$ and a wave drag coefficient, $C_D$. The parameter $\nu_w$ depends on the sea surface temperature (SST) and on the salinity (SAL) and is calculated in accordance with Eqs. (8) and (22) in Sharqawy et al. (2010).

$$W = 3.84 \times 10^{-6} \times u_{10}^{3.41} \tag{1}$$

In the surf zone, the sea salt particle number flux is considerably enhanced compared with that in the open ocean. Kelly et al. (2010) proposed the approach to addressing surf zone emissions that is used in CMAQ, namely, the whitecap coverage $W$ is set to 1 in the surf zone which is assumed to have a width of $50\ \mathrm{m}$. CMAQ simulations of parts of Florida performed well with this definition of the surf zone (Kelly et al., 2010).

### 2.2.3 Technical Implementation

The aerosol particles in CMAQ are represented by particle number, surface area, and mass concentrations (see Sect. 2.1). Therefore, the total particle number, surface area, and mass emissions per size mode must be provided in CMAQ. However, non-sea-salt-particle emissions are read in only as total mass emissions via external input files. These mass emissions are split into the three size modes using pre-defined splitting factors. The number and surface area emissions are calculated on the basis of standardized geometric mean diameters (GMD) and standard deviations for each mode. By contrast, for sea salt emissions in the standard CMAQ setup, all three values are calculated online in the sea salt emission module based on Gong (2003). The parameterization is fitted to two log-normal distributions (Fig. 3), with the GMD, the standard deviation, and the 0th and 3rd moments being prescribed in the sea salt emission module of CMAQ. The number, surface area, and mass emissions are calculated from these prescribed parameters. One of the distributions represents the accumulation mode, and the other represents the coarse mode. For the GO03 emission case, this portion of the implementation was left unchanged. By contrast, the SP13 and OV14 emissions (number, surface area, and mass) were calculated externally and read by CMAQ at run time. The particulate sea salt is emitted only into the bottom layer of the model grid, whereas other emissions might be also emitted into higher grid layers – e.g. because of high stacks or starting planes.

Because OV14 consists of five log-normally distributed modes, the two finest size modes were assigned to the Aitken mode in CMAQ, the third and fourth finest size modes were assigned to the accumulation mode, and the largest size mode was assigned to the coarse mode. Because SP13 is not based on log-normally distributed modes, it was integrated within fixed boundaries to split it into the Aitken, accumulation, and coarse modes. The boundary between the Aitken and accumulation modes was set to $D_{\mathrm{dry}} = 0.1\ \mu\mathrm{m}$, and the boundary between the accumulation and coarse modes was set to the intersection between the accumulation and coarse modes for GO03 ($D_{\mathrm{dry}} \approx 1.5\ \mu\mathrm{m}$), which depends somewhat on the relative humidity (see Sect. S4.1).

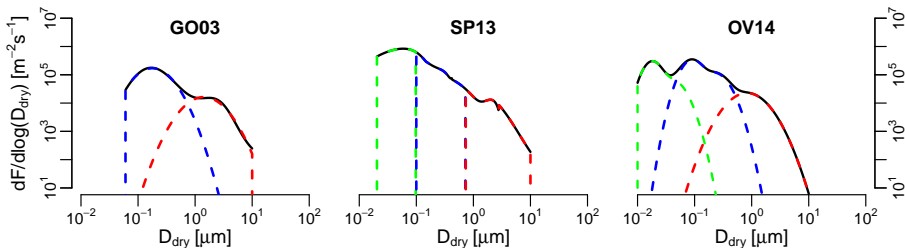

**Figure 3.** Modal splitting of the sea salt emission parameterizations GO03 (left), SP13 (center), and OV14 (right). The color indicates the size mode in which the sea salt is emitted: green corresponds to the Aitken mode, blue to the accumulation mode and red to the coarse mode.

The SAL in the Baltic Sea is very low - below 10‰ throughout large regions - which requires the inclusion of an SAL dependence in the sea salt emission calculation. For GO03, the approach described in Neumann et al. (2016) was applied: number, surface area, and mass emissions were multiplied by $\frac{\text{SAL}}{35‰}$. OV14 already includes salinity as a parameter. For SP13, we added an SAL dependence based on plots published by Mårtensson et al. (2003): the size of the emitted sea salt particles was scaled by $\left(\frac{\text{SAL}}{35‰}\right)^{1/3}$. Graphically, the number emission distribution (Fig. 3 center and Fig. 2 orange line) shifts to the left as the SAL decreases (Fig. S1). The Aitken/accumulation and accumulation/coarse mode integration boundaries were held constant, leading to a decrease in the coarse-mode number emissions with a decreasing SAL. Detailed information on the salinity dependence is provided in the supplement (Sect. S3). The surf zone is treated differently in the three parameterizations. In CMAQ (GO03), the surf zone is treated in accordance with Kelly et al. (2010) by setting the whitecap coverage $W$ to 1 in the surf zone. In this study, calculations of the surf zone size were performed for a $50\,\text{m}$ wide surf zone by ArcGIS, avoiding double-counting of overlapping surf zone stripes (Neumann et al., 2016). The procedure of setting $W$ to 1 can also be applied for SP13 because MA03 and MO86 depend on the same whitecap coverage parameterization as does GO03 (see Sect. S2). However, the SM93 coarse emissions remain unchanged. This approach cannot be applied to OV14 without modification because the wind speed dependence of OV14 is not based on the whitecap coverage approach. Therefore, no surf zone treatment for OV14 was introduced. The total emitted sea salt mass was split into 7.6% $SO_4^{2-}$, 53.9% $Cl^-$, and 38.6% $Na^+$ (Kelly et al., 2010). The $Na^+$ in the model includes $Na^+$, $Mg^{2+}$, $K^+$, and $Ca^{2+}$; only 78% of the $Na^+$ in the model is true $Na^+$. This split was applied for all three parameterizations. In addition to dry sea salt, water is also emitted. For GO03, the water content was calculated according to Zhang et al. (2005), and for SP13 and OV14, it was calculated according to Lewis and Schwartz (2006). Both relations are based on data from Tang et al. (1997). The new sea salt emissions were calculated externally and read at run time by CMAQ. The CMAQ sea salt emission module (SSEMIS.F) was modified for this purpose. In the modified version, sea salt emissions can be calculated internally or read in from an external source. Currently, no Aitken-mode sea salt particles are considered in standard CMAQ. The sea salt emission and aerosol emission modules (AERO_EMIS.F) were modified to consider Aitken-mode sea salt emissions in addition to those considered in the standard implementation. The modified CMAQ modules are attached as supplementary material and briefly documented in Sect. S6.

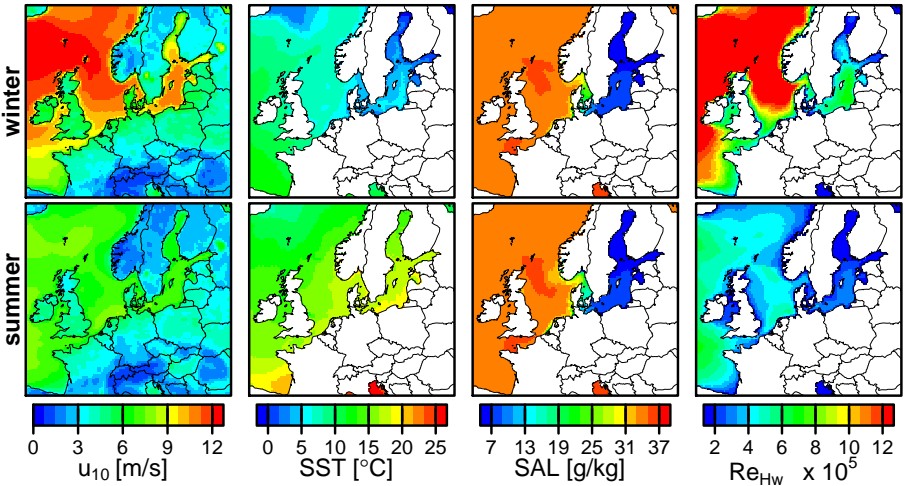

**Figure 4.** Two-month average $u_{10}$, SST, SAL, and $\text{Re}_{\text{Hw}}$ data are plotted for winter (top) and summer (bottom). $\text{Re}_{\text{Hw}}$ was calculated according to Eq. (B7).

## 2.3 Geophysical Input and Emission Data

The land-based emissions were compiled by SMOKE for Europe (Bieser et al., 2011) with the agricultural emissions in accordance with Backes et al. (2016a, b). Dust emissions were not included. Shipping emissions were calculated bottom up using ship movement and ship characteristics data (Aulinger et al., 2016).

The meteorological input data were generated by COSMO-CLM (Consortium for Small-scale Modeling in Climate Mode) (Rockel et al., 2008; Geyer and Rockel, 2013). The $10$ m wind speed is well reproduced above the North Sea (Geyer, 2014; Geyer et al., 2015). The employed data set is part of the coastDatII database of the Helmholtz-Zentrum Geesthacht (Weisse et al., 2015) (http://www.coastdat.de/). The coastDatII database also contains modeled data for wave and ocean currents, which are forced by COSMO-CLM meteorology. The model grid spans the entire model domain. The data were remapped onto the

CMAQ grid, and relevant variables were extracted and converted using a modified version of CMAQ's Meteorology-Chemistry Interface Processor (MCIP) (Otte and Pleim, 2010).

    Wave data ($H_S$ and $u_*$), SAL values, and SST values are required for calculating the new sea salt emissions. For the North Sea, $H_S$ and $u_*$ were obtained from the coastDatII database modeled by the Wave Model (WAM) (Groll et al., 2014). However, Baltic Sea wave data were not available from this database. The significant wave height data for the other seas were acquired

from the ERA-Interim wave data set, which was calculated by WAM for a global domain (Dee et al., 2011). No friction velocity data, $u_*$, were available from that data set; hence, the values of this quantity were calculated from $u_{10}$ (Wu, 1982) using Eqs. (S12) and (S13).

    No SAL and SST fields are present in coastDatII. For the North and Baltic Seas, these data were acquired from operational model runs of the German Federal Maritime and Hydrographic Agency (Bundesamt für Seeschifffahrt und Hydrographie,

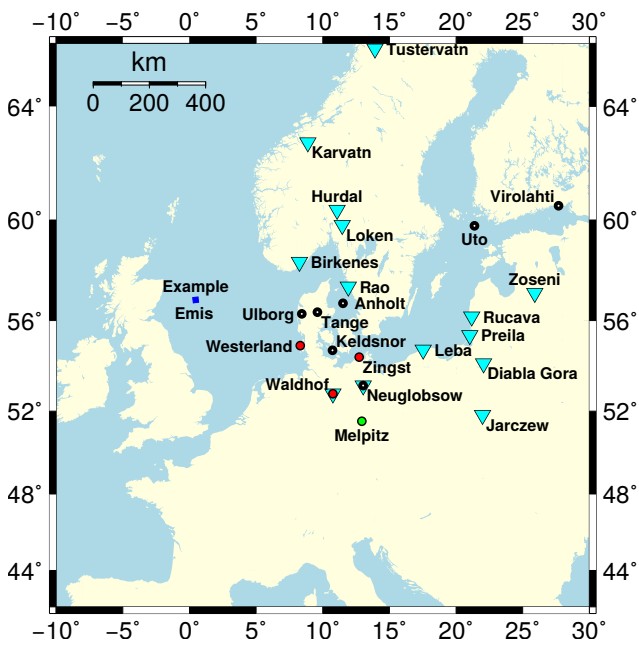

**Figure 5.** Locations of the EMEP stations at which measured and modeled daily average sodium $PM_{10}$ data were compared. Red circles: In addition to statistical data being provided, plots are shown and described in detail. Green circle: An additional comparison of sodium $PM_{2.5}$ data is presented (DE0044R). Blue box: Location of a grid cell which sea salt emissions are presented. Inverted cyan triangles: Sodium wet deposition is evaluated.

BSH) at two different resolutions (see Fig. S2) produced by their model BSHcmod. For the other seas, ERA-Interim SST data were used. The SAL was set to 35‰ in the Atlantic Ocean, 37‰ in the Mediterranean Sea, and 18‰ in the Black Sea.

A detailed listing of the input data sets (Table S6) and their spatial extent (Fig. S2) are presented in the supplement (Sect. S5).

## 2.4   Model Evaluation

5   The model results were compared with atmospheric particulate sodium concentrations measured at 11 EMEP stations, with dissolved sodium in precipitation at 14 EMEP stations and with aerosol optical depth (AOD) measurements at 17 AERONET stations. The EMEP data (Tørseth et al., 2012) were obtained from the EBAS database (http://ebas.nilu.no/). The sodium concentration is an accurate representation of the sea salt concentration because sea salt is the major source of atmospheric sodium and sodium does not evaporate into the gas phase. The AERONET data were obtained from the AERONET homepage

10   (http://aeronet.gsfc.nasa.gov/). Data from winter (January and February) and summer (July and August) of the year 2008 were considered.

### 2.4.1 EMEP

The stations considered in the comparison are listed in Table 2 and plotted in Fig. 5. The last column in Table 2 indicates whether each station is located on the coast or inland (more than 50 km distance to the next coast in the upwind direction). Daily average sodium $PM_{10}$ measurement data are available at all of the stations. In addition, at Melpitz, sodium $PM_{2.5}$

measurements are available and compared against model data. All stations were compared on the basis of statistical parameters: the residual absolute error (RAE), the mean normalized bias (MNB), and Pearson's correlation coefficient (R). The formulas for the RAE, the MNB, and R are given in the appendix as Eqs. (C1) to (C3), respectively. In addition, the data from the Westerland (DE0001R), Waldhof (DE0002R), Zingst (DE0009R) ($Na^+$ in $PM_{10}$, each), and Melpitz ($Na^+$ in $PM_{10}$ and in $PM_{2.5}$) stations were compared graphically. The station of Westerland is located directly at the North Sea coast, the station

of Zingst is located at the Baltic Sea coast, and the station of Waldhof is located approximately 200 km inland. Hence, these stations' measurements cover three different sea salt emission regimes.

For the comparison of model and measurement data, $PM_{10}$, $PM_{2.5}$ and $PM_C$ (= $PM_{10}$ - $PM_{2.5}$) of $Na^+$ were extracted from the model simulation results. Although, particles are represented by three log-normal distributed modes in CMAQ (coarse, accumulation, and Aitken), $PM_C$ does not equal the modeled coarse-mode mass and $PM_{2.5}$ does not equal the sum of Aitken-

and accumulation-mode mass but the modes are actually cut at the given diameters (see Appendix D).

The modeled sodium wet deposition was evaluated using the same statistical metrics as applied on the atmospheric sodium concentrations. Only EMEP stations with a daily measurement interval were considered for the comparison. However, the sampler type differed between the stations: wet only samplers were employed at some stations, whereas bulk samplers were employed at other ones. The sampler type is indicated in Table 2.

### 2.4.2 AERONET AOD

The AERONET stations considered for the comparison are marked in Fig. 6 and listed in Table S9. At each station, the extinction of solar radiation is measured by sun photometers and converted by standardized algorithms to the aerosol optical depth (AOD). Level 2 data were obtained, which implies that the data are quality assured and that cloudy-sky data points are removed. Model data with a liquid water content above $0.01 \text{ kg/m}^2$ were considered as cloudy and were removed.

The model AOD was calculated by integrating extinction coefficients $b_{ext}$ over all vertical model layers. The $b_{ext}$ were calculated with a formula by Pitchford et al. (2007) given in Eq. (2), which is an updated version of an extensively used formula by Malm et al. (1994). Both formulas were derived to calculate ground-level light extinctions from IMPROVE (Interagency Monitoring of Protected Visual Environments) measurements. The employed formula considers the hygroscopic growth of sea salt, ammonium nitrate, and ammonium sulfate particles. Additionally, some particulate compounds are divided into small fine

particulate mass and large fine particulate mass leading to improved results (Pitchford et al., 2007). The "fine" is added because speciated measurements are only performed for $PM_{2.5}$ at the IMPROVE stations. All particles larger than 2.5 μm in diameter are not speciated but considered as bulk particle mass.

**Table 2.** EMEP stations that were considered for comparison with the modeled data. The sampler type for the wet deposition (wet only or bulk) is given in brackets in the data column where applicable. The stations are approximately ordered by their distance downwind to the coast. The stations are divided into three groups by vertical lines: (a) at the coast, (b) inland but considerable influence by marine air, and (c) far inland.

| Station ID | Station Name | Data ($Na^+$ in $PM_x$) | Lon | Lat | Height [m] | Location |
|---|---|---|---|---|---|---|
| DE0001R | Westerland | $PM_{10}$ | 8.31 | 54.93 | 12 | Coast |
| DE0009R | Zingst | $PM_{10}$ | 12.73 | 54.43 | 1 | Coast |
| DK0005R | Keldsnor | $PM_{10}$ | 10.73 | 54.73 | 10 | Coast |
| DK0008R | Anholt | $PM_{10}$ | 11.52 | 56.72 | 40 | Coast |
| SE0014R | Råö | precip (wet only) | 11.91 | 57.39 | 5 | Coast |
| PL0004R | Leba | precip (bulk) | 17.53 | 54.75 | 2 | Coast |
| FI0009R | Utö | $PM_{10}$ | 21.38 | 59.78 | 7 | Coast |
| LT0015R | Preila | precip (wet only) | 21.07 | 55.35 | 5 | Coast |
| LV0010R | Rucava | precip (wet only) | 21.17 | 56.16 | 18 | Coast |
| DK0031R | Ulborg | $PM_{10}$ | 8.43 | 56.28 | 10 | Coast |
| NO0001R | Birkenes | precip (bulk) | 8.25 | 58.38 | 190 | Mixed |
| FI0017R | Virolahti II | $PM_{10}$ | 27.69 | 60.53 | 4 | Coast |
| DK0003R | Tange | $PM_{10}$ | 9.60 | 56.35 | 13 | Inland |
| NO0039R | Kårvatn | precip (bulk) | 8.88 | 62.78 | 210 | Inland |
| NO0015R | Tustervatn | precip (bulk) | 13.92 | 65.83 | 439 | Inland |
| DE0002R | Waldhof | $PM_{10}$, precip (wet only) | 10.76 | 52.80 | 74 | Inland |
| DE0007R | Neuglobsow | $PM_{10}$, precip (wet only) | 13.03 | 53.17 | 62 | Inland |
| LV0016R | Zoseni | precip (wet only) | 25.91 | 57.14 | 188 | Inland |
| PL0005R | Diabla Gora | precip (wet only) | 22.07 | 54.15 | 157 | Inland |
| NO0218R | Løken | precip (bulk) | 11.46 | 59.81 | 135 | Inland |
| NO0056R | Hurdal | precip (bulk) | 11.08 | 60.37 | 300 | Inland |
| DE0044R | Melpitz | $PM_{10}$, $PM_{2.5}$ | 12.93 | 51.53 | 86 | Inland [a] |
| PL0002R | Jarczew | precip (bulk) | 21.98 | 51.82 | 180 | Inland |

[a] located far inland but often influenced by coastal air

$$
\begin{aligned}
b_{\text{ext}} \approx{}& 2.2 \times f_S\left(\text{RH}\right) \times P_{\text{small ammonium sulfate}} + 4.8 \times f_L\left(\text{RH}\right) \times P_{\text{large ammonium sulfate}} \\
& + 2.4 \times f_S\left(\text{RH}\right) \times P_{\text{small ammonium nitrate}} + 5.1 \times f_L\left(\text{RH}\right) \times P_{\text{large ammonium nitrate}} \\
& + 2.8 \times P_{\text{small organic mass}} + 6.1 \times P_{\text{large organic mass}} \\
& + 10.0 \times P_{\text{elemental carbon}} + 1.0 \times P_{\text{fine soil}} \\
& + 1.7 \times f_{SS}\left(\text{RH}\right) \times P_{\text{sea salt}} + 0.6 \times P_{\text{coarse mass}} \\
& + 330 \times \left[\text{NO}_2\left(\text{ppm}\right)\right]
\end{aligned}
\tag{2}
$$

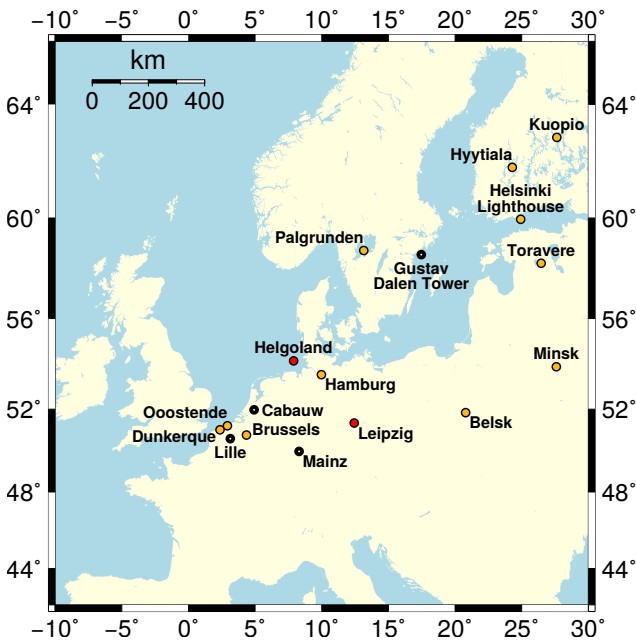

**Figure 6.** Locations of the AERONET stations at which measured and modeled hourly AOD data were compared. Red circles: In addition to statistical data being provided, plots are shown and described in detail. Orange: Statistical metrics are only shown in the Supplement (Table S10).

The $P_i$ denotes the predicted concentration of species $i$ in µg m$^{-3}$. The resulting $b_{\text{ext}}$ has units of Mm$^{-1}$. The total mass of particles $> 2.5\,\mu$m is summarized as bulk mass by $P_{\text{Coarse Mass}}$. For IMPROVE data, the fine particle mass (PM$_{2.5}$) is split into "small" and "large" by empirically derived split factors. Here, the PM$_{2.5}$ mass was split into PM$_{0.1}$ and PM$_{2.5-0.1}$ (= PM$_{2.5}$ - PM$_{0.1}$). The modeled ammonium is divided into ammonium nitrate and ammonium sulfate by the ratio of the negative charges of the nitrate and sulfate masses (see Appendix E for details). The mapping of model species to the $P_i$ is also given in the Appendix.

## 3 Results and Discussion

The first part of this section offers a review of the sea salt emissions produced by the parameterizations. The second part presents a review of the resulting atmospheric concentrations. Finally, the section closes with a summary.

### 3.1 Sea Salt Emissions

In this section, sea salt mass (Sect. 3.1.1), surface area (3.1.2), and number emissions (3.1.3) are described and discussed. The particle surface area is the most important of the three parameters because it governs the impact of the sea salt particles on the atmospheric chemistry: a larger surface area yields a stronger condensation of gases onto sea salt. However, this parameter is

not measured. By contrast, measurements of the speciated particle mass are standardized and available at several measurement stations. Particle number measurements are more complicated to perform, only available at a few stations and not divided into species but given as bulk number concentration. To accurately describe the atmospheric behavior of particle distributions, particle mass, surface area, and number data are needed. Therefore, considering all three types of emissions is relevant.

Figure 7 presents plots of dry sea salt mass emissions. Plots a to f show two-month average dry mass sea salt emissions in winter (left column) and summer (right column) produced with GO03 (1st row), SP13 (2nd row), and OV14 (3rd row). Figure 7 g shows box plots of the sea salt mass emissions split into Aitken, accumulation and coarse modes (left to right) at a location in the German Bight (blue square in Fig. 5) that is representative of the open ocean. Figures 8 and 9 are similar but show sea salt surface area and number emissions, respectively. The time series corresponding to the box plots in the three
figures are given in the supplement (Sect. S8, Figs. S3 to S5).

### 3.1.1   Sea Salt Mass Emissions

The SP13 sea salt mass emissions are considerably higher than those produced by GO03 and OV14. The winter emissions are higher than the summer emissions because of higher wind speeds. The sea salt mass emissions in the Baltic Sea region are quite low because of the SAL scaling. In addition, the difference in emissions between the North and Baltic Seas is partly caused
by differences in wind speed. SP13 emits the most mass per mode, and OV14 emits the least (Fig. 7, a-f). In the coarse and accumulation modes, the GO03 mass emissions lie between those of SP13 and OV14 but closer to the SP13 emissions. The SP13 mass emissions strongly decrease from winter to summer. As indicated in Fig. 2, an additional coarse particle mode exists in SP13 for high wind speeds ($u_{10} > 9 \mathrm{~m~s}^{-1}$). The strong decrease in the SP13 mass emissions in summer originates from a reduced production of spume droplets due to fewer occurrences of threshold exceedance by the wind speed. The coarse-mode
mass emissions are considerably higher than those in the accumulation and Aitken modes. Therefore, they dominate the mass emissions depicted in Fig. 7 g.

### 3.1.2   Sea Salt Surface Area Emissions

In Fig. 8 a-f, the SP13 dry sea salt particle surface area emissions exceed the GO03 and OV14 emissions. However, the GO03 and OV14 surface area emissions are higher than their mass emissions in relation to the respective SP13 emissions. The
surface area emissions are not relevant for the comparisons presented in this study because no measurement data are available. However, they are relevant when considering condensation processes and the formation of $\mathrm{NO_3^-}$, $\mathrm{NH_4^+}$ and $\mathrm{SO_4^{2-}}$. According to Fig. 8 g, the coarse-mode surface area emissions of GO03 and SP13 are close to each other, but those of SP13 are slightly higher. The SP13 accumulation-mode emissions are approximately twice as high as the corresponding GO03 emissions. For all three parameterizations, OV14 produces the lowest emissions in all three modes. The coarse-mode emissions are four to
five times as high as the accumulation-mode emissions and ten to fifty times as high as the Aitken-mode emissions. Hence, the coarse-mode surface area emissions represent the greatest contribution to the total surface area emissions shown in plots a – f.

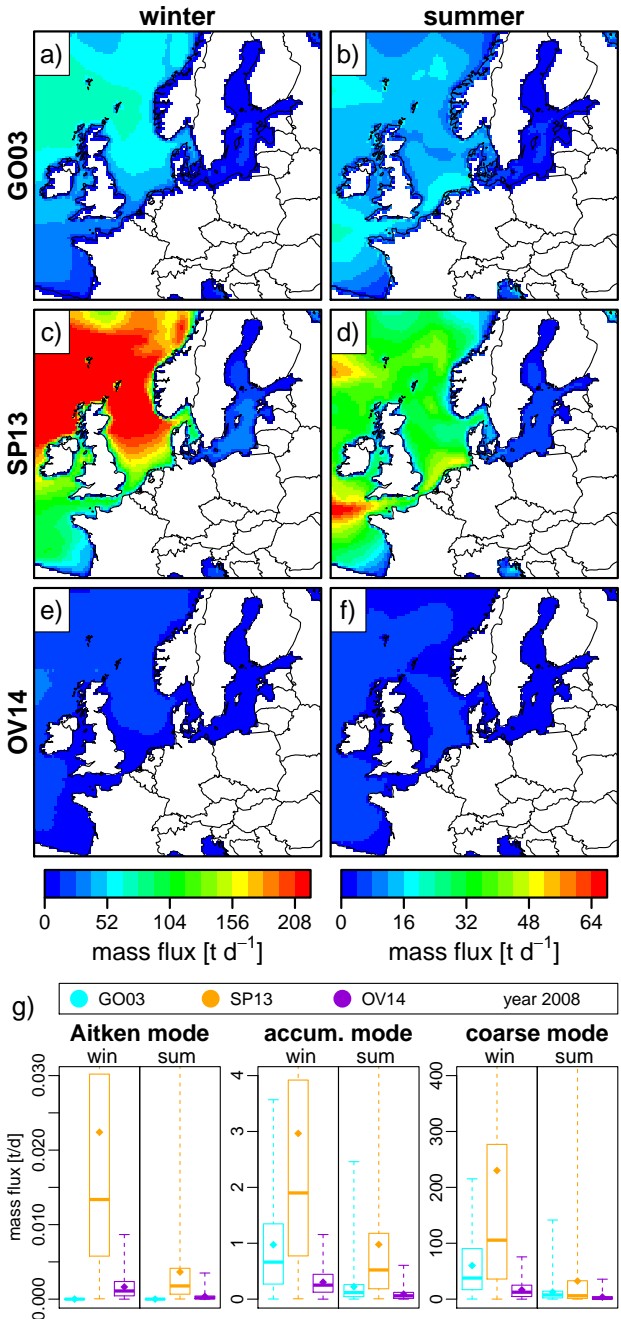

**Figure 7.** Sea salt mass emissions in tons of sea salt per day and grid cell $\left[\mathrm{t\,d}^{-1}\right]$ (total mass of sea salt and not mass of sodium). **a-f**: two-month average mass emissions in winter (left column) and summer (right column). The emissions were calculated using the GO03, SP13, and OV14 (top to bottom) emission parameterizations. The color scale is the same for all plots in the same column. **g**: box plots of mass emissions in the Aitken, accumulation and coarse modes (left to right) at one location in the German Bight (Fig. 5) during summer and winter 2008.

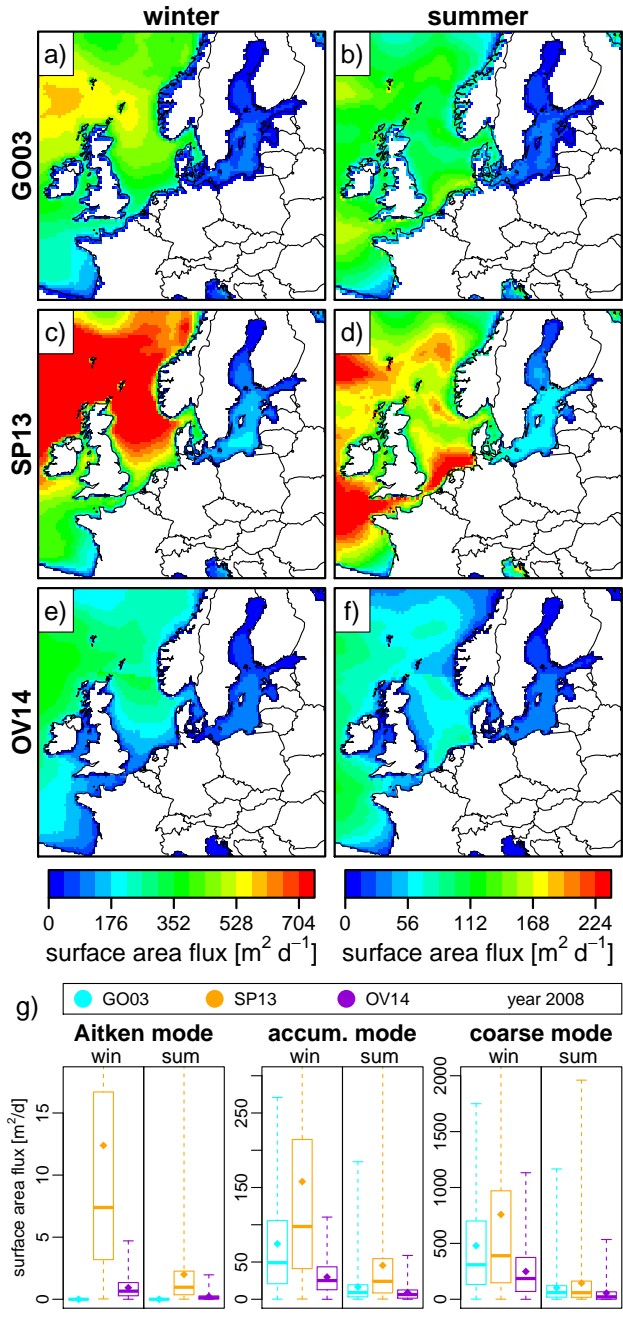

**Figure 8.** Similar to Fig. 7 but showing sea salt surface area emissions. **a-f**: two-month average surface area emissions **g**: box plots of surface area emissions.

### 3.1.3    Sea Salt Number Emissions

The highest total number emissions are calculated using SP13. This is because of the large number of ultra-fine particles on the far left of the distribution in the emission parameterization ($D_{\text{dry}} < 0.1\ \mu\text{m}$), as shown in Fig. 2. For the SP13 parameterization, the relative difference between the Baltic Sea and North Sea number emissions is lower than between the Baltic Sea and North Sea mass emissions. This is because the total mass emissions are scaled by SAL/35‰ and the total number emissions are scaled by 1. Investigation of the modal emissions reveals that the highest coarse-mode number emissions are produced by the OV14 parameterization, followed by GO03. In the accumulation mode, the SP13 number emissions are higher than the corresponding GO03 and OV14 emissions. In the Aitken mode, the SP13 emissions are considerably higher than the respective OV14 emissions. The total number emissions are dominated by the Aitken and accumulation modes. Therefore, SP13 produces the highest total sea salt number emissions, and GO03 produces the lowest. GO03 would probably yield considerably higher particle numbers than OV14 if GO03 included Aitken-mode particles. Because OV14 produces the highest coarse-mode number emissions, one might assume that it also produces the highest coarse-mode surface area and mass emissions. The reason why this is not the case is because the OV14 coarse mode (Fig. 3) consists of particles with a smaller diameter than those in the other two source functions, as confirmed by the GMD (Fig. S5).

### 3.2    Sea Salt Concentrations

#### 3.2.1    Sodium $PM_{10}$ Concentrations

The modeled daily average sodium $PM_{10}$ concentrations were compared with the concentrations measured at 11 EMEP stations. Figure 10 shows the sodium concentrations at three German EMEP stations (Westerland, Waldhof and Zingst) in winter and summer. Table 3 reports the corresponding statistical data for all 11 stations. These stations include both coastal and inland stations (see Table 2), whereas the Melpitz station is located far inland.

At Westerland and Zingst (coastal stations), the SP13 case considerably overestimates the $Na^+$ concentrations and the OV14 case underestimates them. The winter baseline concentrations at Zingst are somewhat well reproduced by all three parameterizations, whereas the highest values (peaks) are not. GO03 overestimates the peak concentrations at Westerland and Zingst. The correlation coefficients for all three parameterizations are close to each other at both stations and in both seasons. However, the MNB is closest to 0 for the OV14 case, followed by GO03 and then SP13. The MNB of OV14 is typically negative, whereas it is positive for the other two cases. The RAE is highest for SP13, and the RAEs of GO03 and OV14 are similar. For all coastal stations in Table 3, the correlation coefficient decreases from winter to summer, whereas the MNBs and RAEs improve except for Westerland. At the station of Westerland in summer, the MNBs and RAEs are highest for SP13. At most coastal stations, the MNBs for the SP13 and GO03 cases are positive and those for the OV14 case are negative. GO03 yielded the lowest RAEs and the MNBs closest to 0 at several of these stations, whereas OV14 yielded the highest RAEs. The latter is caused by high underestimations. The correlation coefficients are quite similar and do not indicate a clear ranking. Notably, at Keldsnor (DK0005R), the correlation coefficients are particularly low.

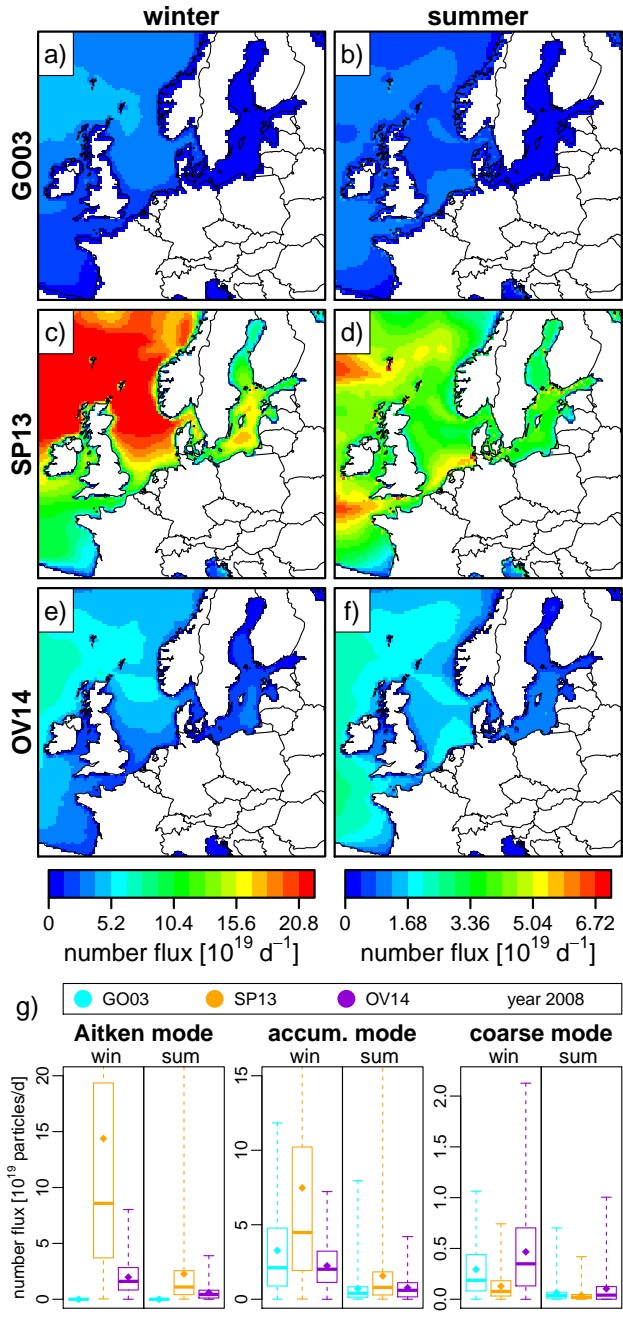

**Figure 9.** Similar to Fig. 7 but showing sea salt number emissions. **a-f**: two-month average number emissions **g**: box plots of number emissions.

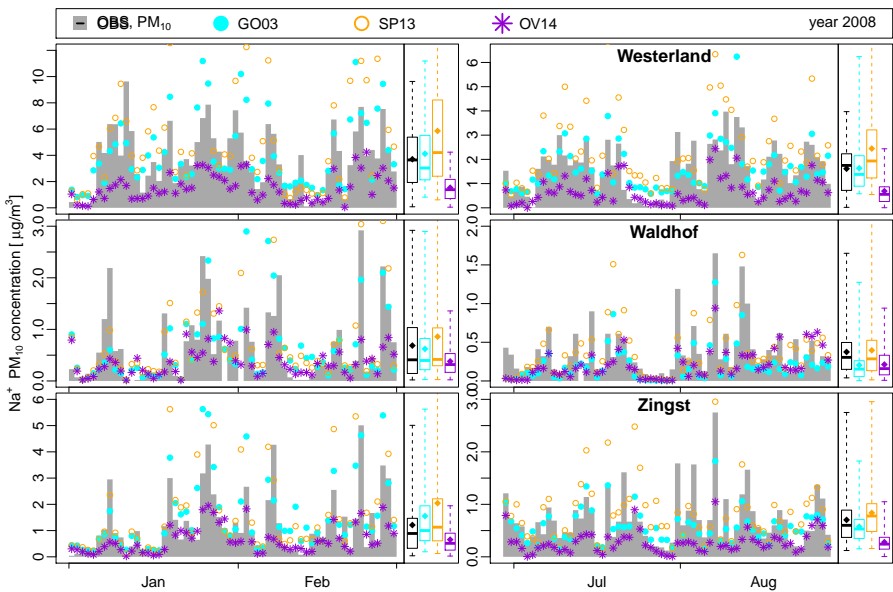

**Figure 10.** Sodium concentrations at three representative EMEP stations (Westerland, Waldhof and Zingst). The black box plot represents the observations. For the box plots of the modeled data, only the daily model values with corresponding measured values are considered.

At Waldhof, which is located approximately $200\,\mathrm{km}$ inland, the modeled concentrations are closer to each other than at the other stations. In winter, SP13 and GO03 overestimate several peak concentrations, but the baseline concentrations are well reproduced by all three parameterizations. In summer, GO03 underestimates the baseline concentration and SP13 appears to yield the best reproduction of the observations. Inland stations exhibit high correlation coefficients of between $0.6$ and $0.8$.

The SP13 emissions yield the highest correlation coefficients. However, their difference to the correlation coefficients of the GO03 and OV14 cases is small. In summer, the inland MNBs of the GO03 and SP13 cases are smaller than those at the coastal stations, indicating less overestimation of the sodium concentrations at inland stations. For the OV14 case, the MNB is positive in approximately half of the inland cases – particularly during winter – whereas it is typically negative at all coastal stations. Thus, OV14 produces fewer underestimations at inland stations. The RAE is often below $0.5$ at inland stations, with

the exception of Tange (DK0003R). Commonly, the winter MNB and RAE values are higher than those in summer. The MNBs and RAEs for Tange deviate most strongly from those for the other stations in this group. Tange is the station that is located closest to the coast. At Melpitz, the MNB of OV14 is positive in both winter and summer. In winter, the MNBs of SP13 and GO03 at Melpitz are lower than those at the other stations. Melpitz is also the station located the furthest from the coast line.

At the coastal station of Keldsnor (DK0005R), the correlation coefficients are very low. During winter, the RAEs are higher

than those at the other stations. The RAEs during summer and the MNBs are in the same range as those at the other stations. Thus, the order of magnitude of the sodium concentrations is well reproduced, whereas the temporal occurrences of the peak concentrations are not well reproduced with respect to the other stations. Keldsnor is located on an island that is not resolved by the model, as is Anholt (DK0008R). However, Anholt is located on a small island that is surrounded only by water, whereas

**Table 3.** Statistical evaluation for the comparisons between the modeled and measured Na$^+$ concentrations at 11 EMEP stations in the vicinity of the North and Baltic Seas during winter (left) and summer 2008 (right).

| sodium PM$_{10}$ | | | Winter 2008 | | | | Summer 2008 | | |
|---|---|---|---|---|---|---|---|---|---|
| Station | Case | $n$ | RAE | MNB | R | $n$ | RAE | MNB | R |
| Westerland | GO03 | 60 | 1.62 | 0.80 | 0.77 | 61 | 0.64 | 1.90 | 0.69 |
| Coast | SP13 | 60 | 2.87 | 1.05 | 0.75 | 61 | 1.09 | 2.79 | 0.71 |
| DE0001R | OV14 | 60 | 2.23 | -0.37 | 0.75 | 61 | 1.01 | -0.12 | 0.70 |
| Zingst | GO03 | 60 | 0.60 | 1.01 | 0.77 | 61 | 0.26 | 0.02 | 0.70 |
| Coast | SP13 | 60 | 1.01 | 1.16 | 0.80 | 61 | 0.36 | 0.47 | 0.59 |
| DE0009R | OV14 | 60 | 0.64 | -0.11 | 0.77 | 61 | 0.43 | -0.57 | 0.76 |
| Keldsnor | GO03 | 60 | 1.09 | 0.59 | 0.45 | 56 | 0.43 | 0.04 | 0.26 |
| Coast | SP13 | 60 | 1.58 | 0.59 | 0.61 | 56 | 0.56 | 0.30 | 0.33 |
| DK0005R | OV14 | 60 | 1.32 | -0.51 | 0.46 | 56 | 0.76 | -0.64 | 0.37 |
| Anholt | GO03 | 59 | 1.02 | 0.35 | 0.81 | 51 | 0.64 | -0.10 | 0.67 |
| Coast | SP13 | 59 | 1.87 | 0.61 | 0.82 | 51 | 0.69 | 0.12 | 0.66 |
| DK0008R | OV14 | 59 | 1.63 | -0.54 | 0.70 | 51 | 1.07 | -0.67 | 0.58 |
| Utö | GO03 | 59 | 0.46 | 1.00 | 0.59 | 61 | 0.26 | 0.09 | 0.66 |
| Coast | SP13 | 59 | 1.12 | 2.07 | 0.65 | 61 | 0.26 | 0.73 | 0.62 |
| FI0009R | OV14 | 59 | 0.32 | 0.21 | 0.61 | 61 | 0.34 | -0.31 | 0.57 |
| Ulborg | GO03 | 60 | 1.14 | 1.44 | 0.74 | 54 | 0.58 | 0.99 | 0.50 |
| Coast | SP13 | 60 | 1.96 | 0.84 | 0.84 | 54 | 0.76 | 0.78 | 0.78 |
| DK0031R | OV14 | 60 | 1.35 | -0.34 | 0.79 | 54 | 0.62 | -0.41 | 0.71 |
| Virolahti II | GO03 | 60 | 0.21 | 1.30 | 0.34 | 54 | 0.10 | 0.05 | 0.74 |
| Coast | SP13 | 60 | 0.35 | 2.08 | 0.45 | 54 | 0.11 | 0.87 | 0.73 |
| FI0017R | OV14 | 60 | 0.18 | 0.72 | 0.33 | 54 | 0.13 | 0.33 | 0.55 |
| Tange | GO03 | 56 | 0.87 | 0.92 | 0.67 | 61 | 0.40 | 0.64 | 0.62 |
| Inland | SP13 | 56 | 1.37 | 1.00 | 0.77 | 61 | 0.58 | 0.93 | 0.73 |
| DK0003R | OV14 | 56 | 0.97 | -0.23 | 0.73 | 61 | 0.45 | -0.32 | 0.67 |
| Waldhof | GO03 | 55 | 0.39 | 1.62 | 0.65 | 60 | 0.20 | -0.42 | 0.70 |
| Inland | SP13 | 55 | 0.47 | 1.80 | 0.73 | 60 | 0.19 | 0.15 | 0.71 |
| DE0002R | OV14 | 55 | 0.40 | 0.63 | 0.73 | 60 | 0.20 | -0.34 | 0.68 |
| Neuglobsow | GO03 | 60 | 0.28 | 1.13 | 0.75 | 59 | 0.19 | -0.45 | 0.72 |
| Inland | SP13 | 60 | 0.36 | 1.22 | 0.83 | 59 | 0.14 | 0.15 | 0.71 |
| DE0007R | OV14 | 60 | 0.36 | 0.41 | 0.75 | 59 | 0.20 | -0.32 | 0.64 |
| Melpitz | GO03 | 59 | 0.25 | 0.30 | 0.66 | 61 | 0.12 | -0.41 | 0.70 |
| Inland | SP13 | 59 | 0.27 | 0.81 | 0.67 | 61 | 0.10 | 0.43 | 0.67 |
| DE0044R | OV14 | 59 | 0.27 | 0.10 | 0.63 | 61 | 0.13 | 0.11 | 0.57 |

Keldsnor is located on a larger island in a region of several islands. Therefore, the local wind fields near Keldsnor may not

be correctly predicted, and consequently, sub-grid deposition processes may not be correctly reproduced by CMAQ, thereby causing the quality of the modeled sea salt concentrations to decline.

The sodium concentrations at coastal stations, such as Westerland and Zingst, are highest for the SP13 emissions and lowest for the OV14 emissions. For locations farther inland, the SP13 and GO03 concentrations decrease more rapidly than the OV14

concentrations, as indicated by the MNBs. At the far-inland station of Melpitz, the SP13 and OV14 cases yield similar sodium concentrations (MNBs, Table 3) that are higher than the GO03 concentrations. In a similar study, Tsyro et al. (2011) also reported slight overestimations at coastal stations and underestimations at some inland stations for GO03 sea salt emissions. In that study, sodium concentrations calculated by the EMEP model were compared with EMEP data of the years 2004 – 2007. Comparing annual average concentrations over all stations yielded overestimations. In contrast, a detailed evaluation

(same study) of sodium $PM_{10}$ and $PM_{2.5}$ data of two EMEP intensive measurement campaigns from June 2006 and January 2007 showed that the model underestimated sodium concentrations in both size fractions at three of four EMEP stations. This result clearly contradicts the results of that study, which clearly highlights the temporal variability of sea salt emissions and indicates that either the emission or the transport processes are not correctly represented by the model. Chen et al. (2016) considerably overestimated sea salt concentrations in WRF-Chem model simulations with GO03 sea salt emissions during

a two-week period in September 2013. The spatio-temporal variation of the concentrations was well captured. Manders et al. (2010) compared particulate sodium concentrations predicted by the LOTOS-EUROS model with EMEP measurements for the year 2005. The sea salt emissions were generated by a combination of the emission parameterizations by Monahan et al. (1986) and Mårtensson et al. (2003), which is similar to the SP13 parameterization. They found that the atmospheric annual mean sea salt concentrations were approximately 2.6-fold overestimated compared to the EMEP measurements. In agreement

with Chen et al. (2016), the spatio-temporal variation was well captured. Moreover, the sea salt concentrations were reduced by 40–50 % when an alternative dry deposition parameterization was employed. Both dry deposition parameterizations in Manders et al. (2010) and the parameterization used in CMAQ are based on the classical resistance approach but the formulations of individual resistances differ. Therefore, a direct comparison of the dry depositions is not possible.

Another important aspect in sea salt modeling studies is the consideration of surf zone emissions: sea salt emissions are

enhance in the surf zone due to increased number of wave breaking events. The generation of surf zone emissions is a complex and small-scale process, which is very difficult to represent in models. Hence, it is commonly not considered in regional scale modeling studies. Kelly et al. (2010) and Gantt et al. (2015) optimized the surf zone emission treatment in CMAQ and found sodium and nitrate concentrations to be better predicted when surf zone emissions were considered. Neumann et al. (2016) identified no improvement of modeled sodium concentrations when surf zone emissions were activated. In this study, the

emissions by the GO03 and SP13 parameterizations incorporate surf zone emissions as suggested by Kelly et al. (2010), where those by OV14 incorporate no surf zone emissions. The different treatment of surf zone emissions might also lead to an offset in the atmospheric sodium concentrations between GO03 and SP13, on the one side, and OV14, on the other side.

The sea salt particle size distributions emitted by the three parameterizations differ considerably as noted in Sect. 3.1. Hence, the atmospheric size distributions are expected to also differ. Because the particle dry deposition is individually calculated

per mode and moment, the sodium concentrations of the three sea salt emission cases are expected to exhibit different dry

deposition velocities. The results in this section indicate that the particle dry deposition velocities for the SP13 and GO03 emission cases are higher than those for OV14 emission case. In the next section (Sect. 3.2.2), the fine ($PM_{2.5}$) and coarse fractions ($PM_C$) of particulate sodium concentrations are evaluated to explain the results of this section in more detail.

### 3.2.2 Particle Size Distribution

In this section, the sea salt particle size distributions in the GO03, SP13, and OV14 cases and their evolution from their source regions toward inland are analyzed. This is performed by considering the $PM_{2.5}$ and $PM_C$ sodium concentrations ($PM_C = PM_{10} - PM_{2.5}$) at the Westerland (coast) and Melpitz (inland) stations. In addition, the modeled $PM_{2.5}$ and $PM_C$ sodium data are compared with measurements at the Melpitz station. This comparison is presented first (Fig. 11) followed by an evaluation of the modeled sodium PM data at both stations (Fig. 12). For more detailed considerations, the $PM_{2.5}$

and $PM_C$ sodium data at Waldhof, which is located at approximately half of the distance between Westerland and Melpitz, and the modeled accumulation and coarse-mode GMDs at the Westerland, Waldhof, and Melpitz stations are provided in the supplement (Figs. S8 and S9).

For the sodium $PM_{2.5}$ concentrations in summer (Fig. 11, center right), GO03 best reproduces the measured concentrations with respect to their magnitude. SP13 and OV14 yield considerable overestimations. During winter, all parameterizations

underestimate the sodium $PM_{2.5}$ peak concentrations, but SP13 overestimates the baseline concentrations, and positive MNBs indicate overestimations in all three cases. The average concentrations are best predicted by OV14, but the MNB is lowest for GO03. The correlation coefficient for OV14 is lower than those for GO03 and SP13 (Table 4). Thus, GO03 produces the best sodium $PM_{2.5}$ predictions, followed by OV14. Because OV14 is based on a highly detailed particle size distribution data set and considers ultra-fine particles (the Aitken mode), it might be expected that this parameterization would yield the best

predictions of the sodium $PM_{2.5}$ particle concentrations.

The temporal occurrences of peak sodium $PM_C$ concentrations are not consistently predicted by the three parameterizations, i.e., GO03 and SP13 predict several peaks that are not predicted by OV14, and OV14 also predicts peaks that are not predicted by the other two cases. The sodium $PM_C$ concentrations are underestimated by all three cases in summer (MNB $< 0$), which leads to underestimation of the sodium $PM_{10}$ concentrations by GO03. OV14 and SP13, in contrast, still moder-

ately overestimate the sodium $PM_{10}$ concentrations because of a considerable overestimation of sodium $PM_{2.5}$. In particular, OV14 considerably overestimates the sodium $PM_C$ concentrations in late August for approximately a week, whereas the other parameterizations predict lower and more accurate concentrations. If this period were to be neglected, a more pronounced negative MNB for OV14 during summer would occur. In winter, the coarse particles are overestimated by all parameterizations (MNB $> 0$); this overestimation is lowest for OV14 and highest for SP13. The correlation coefficients and RAEs for each

season are quite similar and provide no clear indication of which parameterization yields better results. Thus, based on the R values and the RAEs, no parameterization produces a clearly superior prediction of sodium $PM_C$ concentrations. According to the MNBs, OV14 produces slightly better results than the other two cases when winter and summer are considered together.

In summary, GO03 produces the best sodium $PM_{2.5}$ concentrations, and OV14 produces the best sodium $PM_C$ concentrations at Melpitz. This size-resolved comparison indicates that sodium $PM_{10}$ concentrations are not necessarily appropriate for

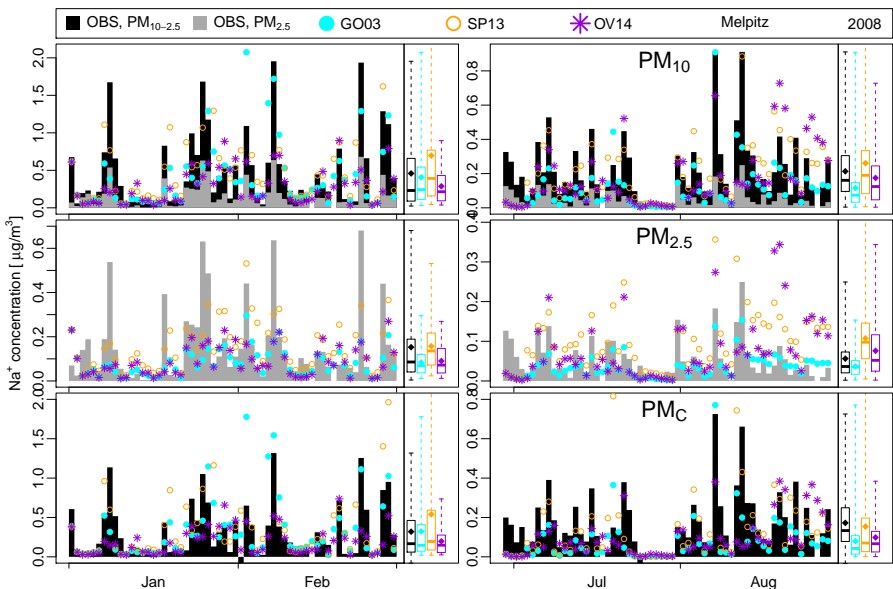

**Figure 11.** Daily average measured and modeled sodium concentrations at the EMEP station at Melpitz. The sodium $PM_{10}$, $PM_{2.5}$ and $PM_C$ concentrations are plotted in the top, center and bottom rows, respectively, for winter (left) and summer (right). The black box plot represents the observations. For the box plots of the modeled data, only the daily model values with corresponding measured values are considered.

**Table 4.** Similar to Table 3 but for the Melpitz station only and for different particle sizes.

| Sodium $PM_x$ | | | winter | | | | summer | | |
|---|---|---|---|---|---|---|---|---|---|
| Size | Case | $n$ | RAE | MNB | R | $n$ | RAE | MNB | R |
| | GO03 | 59 | 0.25 | 0.43 | 0.66 | 61 | 0.11 | -0.35 | 0.69 |
| $PM_{10}$ | SP13 | 59 | 0.39 | 1.27 | 0.67 | 61 | 0.12 | 0.58 | 0.67 |
| | OV14 | 59 | 0.27 | 0.11 | 0.65 | 61 | 0.13 | 0.12 | 0.57 |
| | GO03 | 58 | 0.09 | 0.19 | 0.64 | 56 | 0.03 | 0.08 | 0.50 |
| $PM_{2.5}$ | SP13 | 58 | 0.10 | 1.37 | 0.64 | 56 | 0.07 | 2.28 | 0.45 |
| | OV14 | 58 | 0.10 | 0.39 | 0.52 | 56 | 0.06 | 1.27 | 0.31 |
| | GO03 | 56 | 0.20 | 0.69 | 0.64 | 52 | 0.11 | -0.40 | 0.53 |
| $PM_C$ [a] | SP13 | 56 | 0.35 | 1.42 | 0.65 | 52 | 0.13 | 0.15 | 0.50 |
| | OV14 | 56 | 0.19 | 0.19 | 0.65 | 52 | 0.11 | -0.27 | 0.48 |

[a] Sodium $PM_C$ is calculated as $PM_{10} - PM_{2.5}$ of sodium. In rare situations, $PM_{10} < PM_{2.5}$ exists in the measurements. In these situations, the resulting $PM_C$ value is not considered.

validating sea salt emission parameterizations but that size-resolved measurements are of considerable importance in the validation process. Therefore, size-resolved sodium measurements in coastal regions will be necessary for the further evaluation of sea salt source functions.

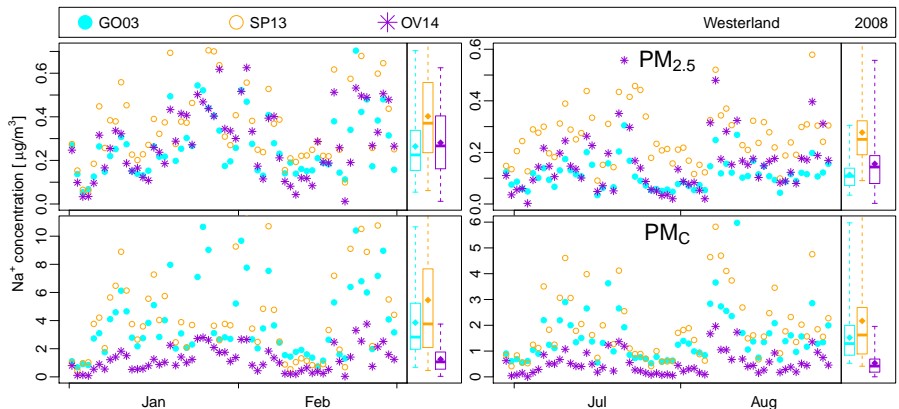

**Figure 12.** Similar to Fig. 11 but showing data for Westerland. No sodium $PM_{2.5}$ data were available and no sodium $PM_C$ concentrations were calculated.

For evaluating the evolution of the sodium size distributions from the coast toward the hinterland, Fig. 12 depicts similar data as Fig. 11 but at the Westerland station. The same plot is presented in Fig. S7 in the supplement for Waldhof, which is located in between Westerland and Melpitz. At Westerland, $PM_C$ sodium represents the predominant contribution to the total sodium mass in all three sea salt emission parameterizations (Fig. 12). The sodium $PM_{2.5}$ and $PM_C$ concentrations are

twice as high during winter than summer. The SP13 case yields the highest sodium $PM_C$ concentrations, and OV14, yields the lowest. By contrast, the OV14 case yields higher sodium $PM_{2.5}$ concentrations than the GO03 case in summer. In winter, the sodium $PM_{2.5}$ concentrations of both cases are on the same level. In contrast at Melpitz, the concentrations are considerably lower than at Westerland, particularly in the SP13 and GO03 cases. The concentrations at Waldhof are in between (Suppl.). The decrease in the sodium $PM_{2.5}$ concentrations from Westerland via Waldhof to Melpitz is lower compared to the decrease

in sodium $PM_C$ concentrations. Therefore, the relevance of the $PM_{2.5}$ sea salt fraction increases with distance to the marine sea salt emission regions. Fine particulate sea salt is more relevant for the transport of species attached to the particles over long distances, such as nitrate. In contrast, coarse sea salt particles are the size fraction predominantly deposing close to their source regions and enhance the deposition flux of attached species, such as nitrogen compounds. Hence, sea salt emission parameterizations that include more fine particles, such as OV14, can be expected to transport higher concentrations of those

attached species over long distances than parameterizations that yield a strong dry deposition close to source regions, such as SP13. The different dry deposition velocities are not only due to a different split of the sea salt mass between accumulation and coarse mode but also due to different GMDs of the modal distributions. Plots of accumulation and coarse-mode GMDs that clearly highlight the differences between the sea salt emission cases are provided in the supplement but are not discussed.

### 3.3 Wet Deposition

Modeled sodium wet deposition of the three sea salt emission cases and sodium wet deposition measurements were compared. Measurements from 14 of more than 30 available stations were chosen for this comparison because the number of measurements per two-month period was above 10 and the stations were not located on high mountains.

The correlation coefficients (Table 5) were below $0.5$ at more than half of the stations throughout the year. In summer, the correlation coefficients at four stations were even negative. The correlation coefficients at the individual stations are closer to each other than those between the stations (in other words, the variation between stations is higher than within one station). None of the three sea salt emission cases clearly yields higher correlation coefficients than the other two.

The MNBs are negative at most stations. They are lowest in OV14 case simulations and highest in SP13 case simulations. Hence at most stations, SP13 and GO03 case simulations are closer to $0$ than the OV14 case simulations. The strong sodium wet deposition underestimations by OV14 correspond to the underestimation in the sodium $PM_{10}$ concentrations. At the stations of Diabla Gora, the MNB of OV14 exceeds that of GO03 which is consistent with the MNBs of the concentrations. In Neumann et al. (2016) with the same CMAQ setup, the nitrate wet deposition was underestimated although atmospheric concentrations of $HNO_3+NO_3^-$ were not. Therefore, it is reasonable to assume that sodium wet deposition should also be underpredicted by CMAQ. Moreover, it is questionable whether the fact, that the MNBs of the SP13 case simulations are closest to $0$ actually indicates that the SP13 parameterization reproduces the real sea salt emissions more accurate than the OV14 parameterization.

The total precipitation amount was also underestimated at some stations (Table S8) but not as strong as the wet deposited sodium mass. However, the temporal variation of the precipitation could not be validated at these stations because the temporal resolution of the measurement data was too low. Comparing the precipitation data against temporally higher resolved measurements at other locations would not replace a validation at the considered EMEP stations because the spatial distribution of small-scale rain showers is very heterogeneous.

Tsyro et al. (2011) also evaluated modeled sodium wet deposition against EMEP data. They found considerable underestimations by more than $50\%$ which is consistent with the result presented above. The amount and temporal resolution were well represented in the meteorological data. Appel et al. (2011) found underestimations of nitrate and ammonium wet deposition using the CMAQ modeling system in the North American region. Although Tsyro et al. (2011) did not use CMAQ but the EMEP model and Appel et al. (2011) and Neumann et al. (2016) did not regard sea salt wet deposition, the consistent results indicate that the wet scavenging might be underestimated by both models.

### 3.4 Aerosol Optical Depth

A visual comparison of AOD data is presented in Fig. 13. AODs at Helgoland (summer, top row) and Leipzip (summer and winter, center and bottom rows, respectively) are plotted. Leipzig is located close to Melpitz and Helgoland is in a similar air quality regime as Westerland. In winter months, no AOD is measured at Helgoland.

**Table 5.** Statistical metrics on modeled and measured sodium wet deposition at 14 EMEP stations. RAE, MNB, R, $\mu_P$ (mean predicted), and $\mu_O$ (mean observed) are shown.

| sodium wet deposition | | | Winter 2008 | | | | | | Summer 2008 | | | | |
|---|---|---|---|---|---|---|---|---|---|---|---|---|---|
| Station | Case | $n$ | RAE | MNB | R | $\mu_P$ | $\mu_O$ | $n$ | RAE | MNB | R | $\mu_P$ | $\mu_O$ |
| Råö | GO03 | 38 | 0.229 | -0.545 | 0.560 | 0.071 | 0.294 | 26 | 0.047 | -0.397 | 0.338 | 0.031 | 0.069 |
| SE0014R | SP13 | 38 | 0.164 | -0.036 | 0.574 | 0.192 | 0.294 | 26 | 0.057 | -0.081 | 0.359 | 0.054 | 0.069 |
| Coast | OV14 | 38 | 0.273 | -0.857 | 0.548 | 0.021 | 0.294 | 26 | 0.060 | -0.812 | 0.359 | 0.010 | 0.069 |
| Leba | GO03 | 31 | 0.028 | 0.568 | 0.374 | 0.024 | 0.037 | 28 | 0.017 | -0.752 | 0.598 | 0.006 | 0.024 |
| PL0004R | SP13 | 31 | 0.045 | 1.907 | 0.410 | 0.056 | 0.037 | 28 | 0.012 | -0.574 | 0.618 | 0.013 | 0.024 |
| Coast | OV14 | 31 | 0.031 | -0.168 | 0.374 | 0.009 | 0.037 | 28 | 0.018 | -0.776 | 0.570 | 0.006 | 0.024 |
| Preila | GO03 | 20 | 0.101 | -0.613 | 0.206 | 0.034 | 0.115 | 29 | 0.065 | -0.318 | -0.170 | 0.006 | 0.067 |
| LT0015R | SP13 | 20 | 0.134 | -0.016 | 0.253 | 0.090 | 0.115 | 29 | 0.066 | 0.436 | -0.152 | 0.014 | 0.067 |
| Coast | OV14 | 20 | 0.100 | -0.825 | 0.217 | 0.015 | 0.115 | 29 | 0.067 | -0.037 | -0.174 | 0.007 | 0.067 |
| Rucava | GO03 | 18 | 0.053 | -0.379 | 0.358 | 0.037 | 0.062 | 30 | 0.020 | -0.666 | 0.254 | 0.003 | 0.024 |
| LV0010R | SP13 | 18 | 0.074 | 0.495 | 0.364 | 0.093 | 0.062 | 30 | 0.018 | -0.408 | 0.281 | 0.007 | 0.024 |
| Coast | OV14 | 18 | 0.046 | -0.689 | 0.391 | 0.017 | 0.062 | 30 | 0.020 | -0.784 | 0.386 | 0.003 | 0.024 |
| Birkenes | GO03 | 37 | 0.250 | -0.714 | 0.399 | 0.074 | 0.324 | 27 | 0.099 | -0.127 | 0.553 | 0.023 | 0.113 |
| NO0001R | SP13 | 37 | 0.210 | -0.284 | 0.395 | 0.214 | 0.324 | 27 | 0.099 | 0.364 | 0.518 | 0.043 | 0.113 |
| Mixed | OV14 | 37 | 0.301 | -0.902 | 0.388 | 0.023 | 0.324 | 27 | 0.105 | -0.685 | 0.499 | 0.009 | 0.113 |
| Kårvatn | GO03 | 31 | 0.191 | -0.750 | 0.161 | 0.007 | 0.197 | 24 | 0.031 | - | -0.336 | 0.025 | 0.022 |
| NO0039R | SP13 | 31 | 0.184 | -0.524 | 0.195 | 0.016 | 0.197 | 24 | 0.033 | - | -0.199 | 0.023 | 0.022 |
| Coast | OV14 | 31 | 0.194 | -0.886 | 0.183 | 0.002 | 0.197 | 24 | 0.020 | - | -0.043 | 0.007 | 0.022 |
| Tustervatn | GO03 | 36 | 0.214 | -0.710 | 0.224 | 0.015 | 0.216 | 22 | 0.011 | - | 0.055 | 0.012 | 0.004 |
| NO0015R | SP13 | 36 | 0.232 | -0.410 | 0.222 | 0.037 | 0.216 | 22 | 0.018 | - | 0.022 | 0.018 | 0.004 |
| Inland | OV14 | 36 | 0.212 | -0.874 | 0.225 | 0.005 | 0.216 | 22 | 0.005 | - | 0.046 | 0.004 | 0.004 |
| Waldhof | GO03 | 19 | 0.025 | 0.652 | 0.375 | 0.025 | 0.028 | 30 | 0.007 | -0.411 | 0.083 | 0.002 | 0.009 |
| DE0002R | SP13 | 19 | 0.050 | 2.491 | 0.411 | 0.055 | 0.028 | 30 | 0.007 | -0.016 | 0.067 | 0.004 | 0.009 |
| Inland | OV14 | 19 | 0.020 | -0.401 | 0.282 | 0.009 | 0.028 | 30 | 0.008 | -0.651 | 0.114 | 0.001 | 0.009 |
| Neuglobsow | GO03 | 22 | 0.010 | 0.212 | 0.546 | 0.016 | 0.016 | 22 | 0.010 | -0.508 | -0.082 | 0.002 | 0.012 |
| DE0007R | SP13 | 22 | 0.025 | 1.609 | 0.532 | 0.035 | 0.016 | 22 | 0.011 | -0.075 | -0.079 | 0.004 | 0.012 |
| Inland | OV14 | 22 | 0.011 | -0.613 | 0.503 | 0.005 | 0.016 | 22 | 0.010 | -0.673 | 0.035 | 0.001 | 0.012 |
| Zoseni | GO03 | 12 | 0.031 | -0.642 | 0.580 | 0.012 | 0.041 | 29 | 0.004 | -0.793 | 0.146 | 0.001 | 0.005 |
| LV0016R | SP13 | 12 | 0.028 | -0.155 | 0.580 | 0.032 | 0.041 | 29 | 0.004 | -0.563 | 0.132 | 0.002 | 0.005 |
| Inland | OV14 | 12 | 0.036 | -0.839 | 0.545 | 0.005 | 0.041 | 29 | 0.004 | -0.776 | 0.148 | 0.001 | 0.005 |
| Diabla Gora | GO03 | 21 | 0.025 | -0.245 | 0.597 | 0.023 | 0.034 | 25 | 0.004 | -0.799 | 0.465 | 0.001 | 0.005 |
| PL0005R | SP13 | 21 | 0.047 | 0.670 | 0.597 | 0.058 | 0.034 | 25 | 0.004 | -0.586 | 0.460 | 0.002 | 0.005 |
| Inland | OV14 | 21 | 0.026 | -0.743 | 0.596 | 0.008 | 0.034 | 25 | 0.004 | -0.818 | 0.511 | 0.001 | 0.005 |
| Løken | GO03 | 32 | 0.029 | -0.146 | 0.425 | 0.017 | 0.034 | 25 | 0.031 | 4.257 | 0.226 | 0.005 | 0.034 |
| NO0218R | SP13 | 32 | 0.039 | 0.572 | 0.454 | 0.037 | 0.034 | 25 | 0.031 | 6.477 | 0.231 | 0.008 | 0.034 |
| Inland | OV14 | 32 | 0.030 | -0.722 | 0.386 | 0.005 | 0.034 | 25 | 0.032 | 0.518 | 0.222 | 0.002 | 0.034 |
| Hurdal | GO03 | 26 | 0.052 | -0.976 | 0.427 | 0.001 | 0.053 | 28 | 0.022 | -0.589 | 0.390 | 0.006 | 0.025 |
| NO0056R | SP13 | 26 | 0.051 | -0.948 | 0.411 | 0.002 | 0.053 | 28 | 0.022 | -0.291 | 0.476 | 0.012 | 0.025 |
| Inland | OV14 | 26 | 0.053 | -0.993 | 0.431 | 0.000 | 0.053 | 28 | 0.022 | -0.840 | 0.397 | 0.002 | 0.025 |
| Jarczew | GO03 | 24 | 0.007 | -0.398 | 0.300 | 0.007 | 0.009 | 17 | 0.004 | -0.919 | -0.296 | 0.000 | 0.004 |
| PL0002R | SP13 | 24 | 0.015 | 0.326 | 0.303 | 0.016 | 0.009 | 17 | 0.004 | -0.827 | -0.296 | 0.001 | 0.004 |
| Inland | OV14 | 24 | 0.006 | -0.765 | 0.270 | 0.003 | 0.009 | 17 | 0.004 | -0.907 | -0.314 | 0.000 | 0.004 |

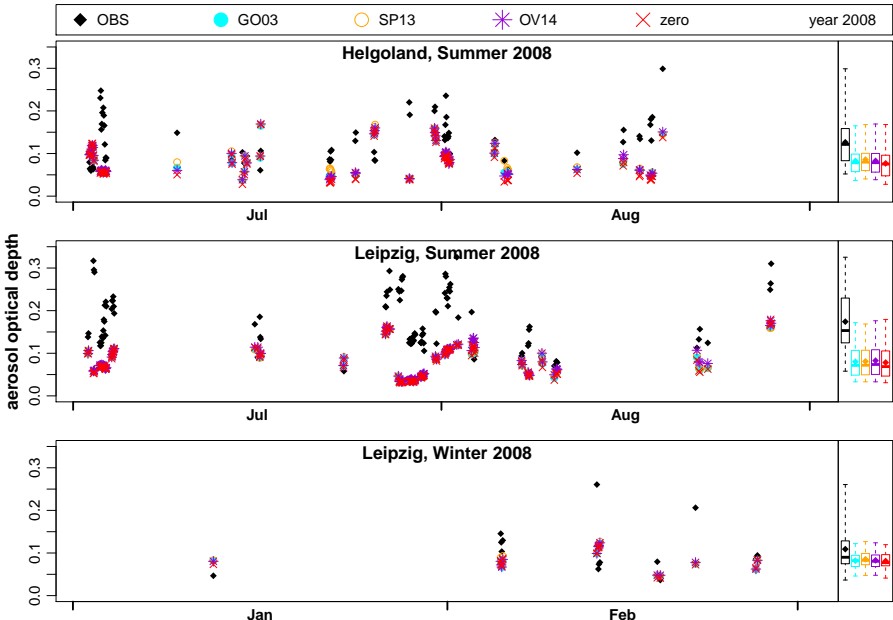

**Figure 13.** Aerosol optical depth (AOD) at the two stations Helgoland (top row) and Leipzig (center and bottom rows). The Helgoland data are only available for summer 2008 and the Leipzig data for summer (center) and winter 2008 (bottom). AODs of measurements (black symbols) and of the four model cases (colored symbols) are plotted. Data points are only plotted if valid model and measurement data are available.

No continuous time series of data points are available because only time points with cloudy sky were dropped. The difference between modeled and measured AODs is commonly larger than the variation between the four model cases. Additionally, there is no clear pattern among the model cases, which indicates that one case yielded AODs closer to the measurements.

The statistical metrics (Table 6) lead to similar conclusions: the correlation coefficients, RAEs, and MNBs of the four model

5 cases are close to each other but no case yields "better" metrics than the other cases. The correlation coefficients are very low – above $0.5$ only in summer 2008 at Mainz – and even negative in three situations. The AODs are underestimated at most stations in summer and at approximately half of the stations in winter (Tables 6 and S10).

Matthias (2008) and Matthias et al. (2012) evaluated model-derived AODs with measurements. They also used the CMAQ model but another CMAQ version and the AOD formula of Malm et al. (1994), which does not include sea salt particles. They

10 found the AOD to be underestimated. This result agrees with this study's negative MNBs in summer 2008. A reason for the too low AODs might be too low or missing biogenic emissions of organic compounds: the formation of secondary organic aerosols (SOA), which considerably impact the AOD, was reduced by a lack of gaseous organic compounds (Matthias, 2008). Additionally, the missing consideration of particulate sea salt in the parameterization of Malm et al. (1994) could have led to the underestimations in the previous studies. Although a more recent AOD formula that includes sea salt mass was used in

this study, AOD is still underestimated. Hence, SOA formation and other primary or secondary particle sources might still be missing in the model setup.

**Table 6.** Statistical evaluation for the comparisons between the modeled and measured aerosol optical depth (AOD) at 6 AERONET stations in the vicinity of the North and Baltic Seas during winter (left) and summer 2008 (right). The data on the remaining 11 AERONET stations are shown in Table S10

| aerosol optical depth | | | Winter 2008 | | | | Summer 2008 | | |
|---|---|---|---|---|---|---|---|---|---|
| Station | Case | $n$ | RAE | MNB | R | $n$ | RAE | MNB | R |
| Helgoland | GO03 | 0 | - | - | - | 75 | 0.06 | -0.21 | -0.10 |
| Coast | SP13 | 0 | - | - | - | 75 | 0.06 | -0.18 | -0.23 |
| | OV14 | 0 | - | - | - | 75 | 0.06 | -0.22 | -0.07 |
| | ZERO | 0 | - | - | - | 75 | 0.07 | -0.27 | -0.06 |
| Cabauw | GO03 | 67 | 0.05 | 0.81 | 0.25 | 81 | 0.08 | 0.00 | -0.29 |
| Coast | SP13 | 67 | 0.06 | 0.90 | 0.28 | 81 | 0.08 | 0.03 | -0.33 |
| | OV14 | 67 | 0.06 | 0.84 | 0.25 | 81 | 0.08 | 0.03 | -0.29 |
| | ZERO | 67 | 0.05 | 0.77 | 0.21 | 81 | 0.08 | 0.00 | -0.27 |
| Lille | GO03 | 93 | 0.05 | 0.49 | -0.13 | 76 | 0.07 | -0.15 | 0.49 |
| Coast | SP13 | 93 | 0.05 | 0.57 | -0.16 | 76 | 0.07 | -0.10 | 0.45 |
| | OV14 | 93 | 0.05 | 0.51 | -0.14 | 76 | 0.07 | -0.12 | 0.50 |
| | ZERO | 93 | 0.06 | 0.54 | -0.13 | 76 | 0.08 | -0.18 | 0.47 |
| Gustav Dalen Tower | GO03 | 0 | - | - | - | 217 | 0.04 | -0.05 | 0.15 |
| Mixed | SP13 | 0 | - | - | - | 217 | 0.04 | -0.02 | 0.16 |
| | OV14 | 0 | - | - | - | 217 | 0.04 | -0.02 | 0.13 |
| | ZERO | 0 | - | - | - | 217 | 0.04 | -0.12 | 0.11 |
| Mainz | GO03 | 96 | 0.05 | 0.08 | 0.55 | 98 | 0.08 | -0.13 | 0.26 |
| Inland | SP13 | 96 | 0.05 | 0.15 | 0.57 | 98 | 0.08 | -0.12 | 0.26 |
| | OV14 | 96 | 0.05 | 0.09 | 0.55 | 98 | 0.08 | -0.11 | 0.25 |
| | ZERO | 96 | 0.05 | 0.10 | 0.48 | 98 | 0.08 | -0.14 | 0.24 |
| Leipzig | GO03 | 14 | 0.05 | -0.05 | 0.03 | 103 | 0.10 | -0.49 | 0.26 |
| Inland | SP13 | 14 | 0.05 | -0.01 | 0.07 | 103 | 0.10 | -0.48 | 0.23 |
| | OV14 | 14 | 0.05 | -0.04 | 0.02 | 103 | 0.09 | -0.47 | 0.22 |
| | ZERO | 14 | 0.05 | -0.07 | 0.09 | 103 | 0.10 | -0.50 | 0.27 |

To summarize, the impact of the sea salt emission parameterization on the modeled AOD is very low and considerably lower than the deviation of the model cases' from the measured AODs. Even AODs at Helgoland, which is clearly dominated by marine air, are only slightly impacted. Therefore, the comparison of AOD data brings no new insights into the evaluation of the three sea salt emission parameterizations. Moreover, the comparison of modeled and measured AODs clearly shows that vertical particle concentrations are still not sufficiently well represented by models or that the AODs are not properly calculated by the used formula.

## 3.5 General Discussion

In this section, the shortcomings of and possible improvements to the individual sea salt emission parameterizations are discussed. The last paragraph contains technical remarks on the sea salt emission calculations.

Because the SP13 sea salt mass concentrations often considerably exceed the measured sea salt concentrations, it can be
assumed that the SP13 emissions are too high. SP13 is based on a laboratory study (Mårtensson et al., 2003) in which SST-dependent sea salt emissions were measured directly after formation. The particle flux measured in Mårtensson et al. (2003) was the gross particle flux, which is not necessarily equal to the net particle flux because some particles fall back into the ocean shortly after their emission. This may explain why SP13 overestimates sea salt emissions. The gross emission flux distribution of Mårtensson might need to be corrected by a size-dependent scaling function to accurately represent the net particle flux.
The development of such a scaling function is beyond the scope of this study. Alternatively, the spume droplet production contributed by SM93, which is activated for wind speeds above a threshold of $9 \mathrm{~m~s^{-1}}$, might be too high. This criterion is exceeded more frequently during winter than during summer. This might yield the higher overestimations at coastal stations during winter compared with those during summer, which were observed in Sect. 3.2. Note that the sodium concentrations would have been considerably stronger overestimated if modeled total suspended particulate matter (TSP) of sodium (not
shown) were compared with EMEP sodium $PM_{10}$ measurements rather than the modeled sodium $PM_{10}$ concentrations as done in this study. Therefore, it is important to consider modeled $PM_{10}$ and not modeled TSP values.

The elevated overestimation at coastal stations during winter has also been observed in the GO03 case. Because both parameterizations depend on the same whitecap coverage parameterization (Monahan and Muircheartaigh, 1980), the increased overestimation during winter might originate from this whitecap coverage parameterization. Massel (2007b) discussed the sen-
sitivity of the exponent in the whitecap coverage parameterization (Eq. (1)). A lower exponent would reduce the gradient of $W(u_{10})$ and the overestimation at high wind speeds. Additionally, GO03 does not include an SST dependence. As described by Mårtensson et al. (2003), Callaghan et al. (2014), and Salter et al. (2015), sea salt emissions decrease with decreasing SST. Thus, an emission reduction in winter due to a low SST might be missing from this model. Using CMAQ version 5.1, different modifications of the GO03 parameterization were compared. Among others, an SST scaling of GO03 emissions reported by
Jaeglé et al. (2011) was tested and found to improve the modeled sodium concentrations. Therefore, it is unclear whether the classical whitecap coverage dependence or deficiencies in the wind-independent part of the parameterization are responsible for the greater overestimation observed during winter.

In contrast to the GO03 and SP13 emission cases, the OV14 case yielded underestimations of the sodium $PM_{10}$ concentrations at coastal stations. OV14 was fitted to data from the Northeastern Atlantic Ocean and to measurements from Mace Head,
Ireland. The Atlantic Ocean is a deep and open ocean, in contrast to the North Sea, which is constrained by several coasts and is quite shallow in most areas. This allows waves to evolve differently; for example, the significant wave height is reduced near Dogger Bank. Hence, it might be necessary to refit the OV14 parameterization to the wave regime in the North Sea, e.g., by scaling $Re_{Hw}$ with the wave period or wave length. An alternative approach that utilizes wave data is based on the energy dissipation caused by wave breaking, as reported by Long et al. (2011). These authors related the volume of air entrained

into the water via wave breaking to the dissipated energy. The volume of entrained air is considered to be proportional to the number of bursting bubbles and the number of sea salt particles produced. Salter et al. (2015) also employed this approach. However, Long et al. (2011) calculated the dissipated energy from $u_{10}$ using a power-law relation, which is simply another fit similar to (Eq. (1)) and does not solve the problem of breaking waves in shallow water. Wave models can also be used to calculate dissipative energy. However, these estimations are rough because no dissipative energy measurements are available for validation purposes (Massel, 2007a).

Surf zone emissions are not the focus of this study. However, they must be briefly discussed because the three compared sea salt emission parameterizations allow the surf zone to be considered in different ways. The wind speed dependence adopted in GO03 and SP13 is the classical Monahan whitecap coverage parameterization (Monahan and Muircheartaigh, 1980). Therefore, the CMAQ surf zone approach described by Kelly et al. (2010), namely, a $50 \text{ m}$ wide surf zone in which the whitecap coverage is set to 1, was applied for these two parameterizations. However, OV14 does not incorporate the classical Monahan whitecap coverage treatment. Rather, a Reynolds number (Eq. B7) is calculated for the sea surface and input into power laws for scaling the five log-normal particle number distributions. Unfortunately, the Reynolds number decreases toward the coast as a result of the decreasing wind speed and decreasing significant wave height (Fig. 4), which leads to reduced OV14 emissions at the coastline. Thus, the OV14 emissions are reduced in the surf zone, in contrast to the increase in surf zone emissions produced by the two other parameterizations. This may be a second reason for why OV14 underestimates the sodium mass concentrations at coastal EMEP stations. An alternative approach that is instead based on the dissipative energy by wave breaking would imply enhanced sea salt emissions in the surf zone and would render a special treatment of the surf zone unnecessary.

The splitting of sea salt emissions into the three aerosol modes is a relevant step that affects the CTM calculations. According to Fig. 2, more particles larger than $2.5 \text{ µm}$ are produced by the SP13 parameterization than by the other two. However, the modal split is different for all three parameterizations (Fig. 3), leading to the emission of smaller but more numerous coarse-mode particles in the OV14 parameterization compared with the others. Consequently, the derived GMD for the OV14 coarse-mode emissions is smaller than those for the SP13 and GO03 coarse-mode emissions (Fig. S5). This affects the modal distribution of the atmospheric particle concentrations (Figs. 11 to 12 and S7 to S9) and atmospheric processes such as dry deposition. Therefore, the technical aspects of the progression from the emission parameterization to the CTM affect the modeled sea salt particle behavior.

## 4   Conclusions

In a comparison of the sodium concentrations produced by three sea salt source parameterizations, the GO03 and OV14 parameterizations were identified as producing sodium mass concentrations closest to measurements. When comparing the modeled sodium $PM_{10}$ mass concentrations to observations, the correlation coefficients in all three cases are often close to each other at individual stations and reveal no overall tendency (Table 3). The MNBs and RAEs indicate that the GO03 and OV14 parameterizations reproduce the measured data better than does the SP13 parameterization, which has the highest MNBs and generally overestimates the sodium concentrations. At coastal stations, OV14 underestimates and GO03 overestimates the sodium con-

centrations, whereas at inland stations, OV14 in general overestimates and GO03 in general underestimates (Fig. 10). This opposite trend between coastal and inland stations is due to the different dry deposition velocities of the parameterizations originating from their different particle size distributions. Considering sodium measurements in the $PM_{2.5}$ and $PM_{10}$ fractions from the Melpitz station, the three parameterizations reproduce the sodium concentrations in these two size classes with vary-

ing degrees of success: GO03 best reproduces the sodium $PM_{2.5}$ mass concentrations, and OV14 best reproduces the sodium $PM_C$ mass concentrations. Unfortunately, no further size-resolved data were available, although measurements from closer to the coast would have been more informative. However, these results clearly indicate that size-resolved measurements are necessary for validating sea salt emission parameterizations.

  The consideration of correlation coefficients and errors of the comparison between modeled and measured sodium wet

deposition did not allow a ranking of the three sea salt emission cases. In contrast, the MNBs identified the SP13 case as yielding sodium wet deposition closest to measurements: the sodium wet deposition was underestimated at most stations and it was least underestimated by the SP13 case. However, other studies suggest that sodium wet deposition might be generally underestimated. It is unclear how strong the sodium wet deposition is underestimated. Therefore, the results of wet deposition evaluation are not clear to interpret.

The three sea salt emission cases only induce a small deviation in the modeled AODs that is considerably lower than the difference of the model cases to the measured AODs. Thus, the comparison with AERONET data does not reveal new insights with respect to the assessment of the three sea salt emission parameterizations. The comparison of spatially resolved modeled and satellite-derived AODs might yield further findings. However, satellite data have a very coarse temporal resolution and the comparison of spatially resolved data needs an entirely new set of statistical metrics, which are beyond the scope of this article.

Moreover, the deviation of the AOD between the sea salt emission scenarios is very low. Hence, also the spatial pattern of AOD is expected to show litter to no deviations between the emission cases. Particularly in summer 2008, AODs were underestimated by the model. The reasons for this might be a too low formation of secondary organic aerosols or an inappropriate choice of the formula for the AOD calculation from model data.

  The GO03 and OV14 emissions yielded the most accurate sodium mass concentrations. However, both parameterizations

have certain shortcomings, and improvements to them should be considered. Enhancing GO03 by SST dependence, such as Jaeglé et al. (2011) did, might reduce overestimations, particularly during winter. OV14 was fitted based on wave data from the Northeast Atlantic Ocean to sea salt measurement data recorded at Mace Head, Ireland. However, the wave spectrum in the Atlantic Ocean is different from that in the North Sea; on the one hand, it may require a refit of the OV14 parameterization to the wave spectrum in the study region. Additionally, the possibility of enhancing OV14 with an appropriate representation of

surf zone emissions should be considered. On the other hand, considering dissipative energy by wave breaking rather than a Reynolds number of the sea surface would probably solve the surf zone and wave spectrum issues.

  Two two-month periods in winter and summer 2008 were evaluated in this study. The results of Tsyro et al. (2011) clearly showed that the model skill to predict atmospheric sea salt concentrations varies throughout the year and between different years. Therefore, one needs to be careful in generalizing the conclusions obtained by this study. Moreover, the processing

of aerosol particles and their vertical transport – particularly, the dry and wet deposition parameterizations – are important

**Table 7.** Parameters, their units and their meaning.

| Parameter | Unit | Explanation |
|---|---|---|
| $r_{80}$ | µm | particle radius at 80% relative humidity |
| $D_{\text{dry}}$ | µm | dry particle diameter |
| $PM_{10}$ | µg m$^{-3}$ | fine and coarse particle ($\leq 10$ µm) mass |
| $PM_{2.5}$ | µg m$^{-3}$ | fine particle ($\leq 2.5$ µm) mass, $\neq \sum$ CMAQ Aitken- and accumulation-mode mass |
| $PM_C$ | µg m$^{-3}$ | coarse particle mass: $PM_{10} - PM_{2.5}$, $\neq$ CMAQ coarse-mode mass |
| $u_{10}$ | m s$^{-1}$ | 10 m wind speed |
| SST | K | sea surface temperature |
| SAL | ‰ | sea surface salinity |
| W | - | whitecap coverage between 0 (0%) and 1 (100%) |
| $u_*$ | m s$^{-1}$ | friction velocity at the sea surface |
| $H_S$ | m | significant wave height |
| $C_D$ | - | drag coefficient due to wind waves |
| $\nu_W$ | m$^2$ s$^{-1}$ | sea water kinetic viscosity |
| Re$_{\text{Hw}}$ | - | Reynolds number of the sea surface due to waves |
| RH | % | relative humidity |
| GMD | µm | geometric mean diameter |
| $\sigma$ | - | standard deviation |
| $\frac{dF}{dr_{80}}, \frac{dF}{dD_{\text{dry}}}$ | $\frac{\text{number}}{\text{m}^2 \text{ µm s}}$ | particle number flux |
| $\frac{dF}{d\log D_{\text{dry}}}$ | $\frac{\text{number}}{\text{m}^2 \text{ s}}$ | particle number flux |
| $\rho_{\text{ss}}$ | g cm$^{-3}$ | density of dry sea salt |

factors affecting the outcome of model study like this one. Hence, it is of high importance to assess eligible sea salt emissions parameterizations over longer time scales and in different chemistry transport modeling systems.

## Appendix A: Abbreviations

Table 7 shows the numbers and meaning of all abbreviations and variables used in the manuscript and in the supplement.

## Appendix B: Sea Salt Emission Parameterizations

### B1  GO03

The sea salt emission parameterization GO03 reported by Gong (2003) is given by Eq. (B1).

$$
\begin{aligned}
\frac{dF_{\text{GO03}}}{dr_{80}} &= W \times 3.576 \times 10^5 r_{80}^{-A} \left(1 + 0.057 \times r_{80}^{3.45}\right) \times 10^{1.607 e^{-B^2}} \\
&= 1.373 \times u_{10}^{3.41} r_{80}^{-A} \left(1 + 0.057 \times r_{80}^{3.45}\right) \times 10^{1.607 e^{-B^2}} \\
A &= 4.7 \times \left(1 + \Theta \times r_{80}\right)^{-0.017 \times r_{80}^{-1.44}} \\
B &= \left(0.433 - \log_{10} r_{80}\right)/0.433 \\
\Theta &= 30
\end{aligned}
\tag{B1}
$$

The parameterization is valid on the size range $0.07\ \mu\text{m} \leq r_{80} \leq 20\ \mu\text{m}$.

### B2  SP13

The parameterization SP13 reported by Spada et al. (2013) consists of MO86, SM93, and MA03. Below, all three formulas are given in Eqs. (B2), (B3), and (B4), respectively. Equation (B5) defines the combination of all three parameterizations.

$$
\begin{aligned}
\frac{dF_{\text{MO86}}}{dr_{80}} &= W \times 3.576 \times 10^5 r_{80}^{-3} 10^{1.19 e^{-B^2}} \\
&= 1.373 \times u_{10}^{3.41} r_{80}^{-3} 10^{1.19 e^{-B^2}} \\
B &= \left(0.380 - \log_{10} r_{80}\right)/0.650
\end{aligned}
\tag{B2}
$$

The parameterization is valid on the size range $0.8\ \mu\text{m} \leq r_{80} \leq 20\ \mu\text{m}$.

$$
\frac{dF_{\text{SM93}}}{dr_{80}} = \sum_{k=1}^{2} \left( A_k\left(u_{10}\right) \times \exp\left(-f_k \left(\ln\left(\frac{r_{80}}{r_{0k}}\right)\right)^2\right)\right)
\tag{B3}
$$

$$
\begin{aligned}
\log_{10} A_1 &= 0.0676 \times u_{10} + 2.43 \\
\log_{10} A_2 &= 0.959 \times u_{10}^{0.5} - 1.476 \\
r_{01} &= 2.1\ \mu\text{m}; \quad r_{02} = 9.2\ \mu\text{m} \\
f_1 &= 3.1; \quad f_2 = 3.3
\end{aligned}
$$

Spada et al. (2013) considers the parameterization to be valid on the size range $5\ \mu\text{m} \leq r_{80} \leq 30\ \mu\text{m}$.

$$\frac{dF_{\text{MA03}}}{dD_{\text{dry}}} = W \times (A \times \text{SST} + B) \tag{B4}$$

$$A = c_4 \times D_{\text{dry}}^4 + c_3 \times D_{\text{dry}}^3 + c_2 \times D_{\text{dry}}^2 + c_1 \times D_{\text{dry}} + c_0$$

$$B = d_4 \times D_{\text{dry}}^4 + d_3 \times D_{\text{dry}}^3 + d_2 \times D_{\text{dry}}^2 + d_1 \times D_{\text{dry}} + d_0$$

The parameterization is valid on the size range $0.02\ \mu\text{m} \leq r_{80} \leq 2.8\ \mu\text{m}$.

$$\frac{dF_{\text{SP13}}}{dD_{\text{dry}}} = \begin{cases} \frac{dF_{\text{MA03}}}{dD_{\text{dry}}} & D_{\text{dry}} \leq 2.8\ \mu\text{m} \\ \frac{dF_{\text{MO86}}}{dD_{\text{dry}}} & D_{\text{dry}} > 2.8\ \mu\text{m} \\ & \wedge u_{10} < 9\ \text{m s}^{-1} \\ \max\left(\frac{dF_{\text{MO86}}}{dD_{\text{dry}}}, \frac{dF_{\text{SM93}}}{dD_{\text{dry}}}\right) & D_{\text{dry}} > 2.8\ \mu\text{m} \\ & \wedge u_{10} \geq 9\ \text{m s}^{-1} \end{cases} \tag{B5}$$

SP13 is valid on the size range $0.02\ \mu\text{m} \leq D_{\text{dry}} \leq 30\ \mu\text{m}$. The parameters $c_i$ and $d_i$ are provided in Table S1.

## B3  OV14

The sea salt emission parameterization OV14 reported by Ovadnevaite et al. (2014) is given by Eq. (B6).

$$\frac{dF_{\text{OV14}}}{d\log_{10} D_{\text{dry}}} = \sum_{i=1}^{5} \frac{F_i\left(\text{Re}_{\text{Hw}}\right)}{\sqrt{2\pi} \times \log_{10} \sigma_i} \times \exp\left(-\frac{1}{2}\left(\frac{\log_{10} \frac{D_{\text{dry}}}{\text{GMD}_i}}{\log_{10} \sigma_i}\right)\right) \tag{B6}$$

$$\text{Re}_{\text{Hw}} = \frac{u_* \times H_S}{\nu_W} = \frac{\sqrt{C_D} \times u_{10} \times H_S}{\nu_W} \tag{B7}$$

The kinetic viscosity $\nu_W$ is calculated according to Eqs. (22) and (8) in Sharqawy et al. (2010). The source function is valid on the size range $0.015\ \mu\text{m} < D_{\text{dry}} < 6\ \mu\text{m}$. The values for $\text{GMD}_i$, $\sigma_i$, and $F_i$ are given in the supplement (Table S2) and in Ovadnevaite et al. (2014).

## Appendix C:  Statistical Evaluation

The statistical figures residual absolute error (RAE), mean normalized bias (MNB), and Pearson's correlation coefficient (R) are calculated according to Eqs. (C1), (C2), and (C3), respectively.

$$\text{RAE} = \frac{1}{n} \times \sum_{i=1}^{n} |P_i - O_i| \tag{C1}$$

$$\text{MNB} = \frac{1}{n} \times \sum_{i=1}^{n} \frac{P_i - O_i}{O_i} \tag{C2}$$

$$\text{R} = 1 - \frac{6}{n(n^2-1)} \times \sum_{i=1}^{n} (P_i - O_i)^2 \tag{C3}$$

with

$P_i$     i$^{\text{th}}$ predicted value

$p_i$     rank of the i$^{\text{th}}$ predicted value

5     $O_i$     i$^{\text{th}}$ observed value

$o_i$     rank of the i$^{\text{th}}$ observed value

$n$     number of observations

## Appendix D: Calculating $\text{PM}_\text{x}$

In CMAQ, the particle mass is represented by three log-normal distributed modes denoted as I, J, and K modes – Aitken, accumulation, and coarse mode, respectively. The $\text{PM}_\text{x}$ masses of the individual modes need to be obtained and summed to obtain the $\text{PM}_\text{x}$ mass over all modes. The latter can be calculated by the modal mass fractions $f_{m,x}$:

$$f_{m,x} = \frac{\text{Mass}_m(x)}{\text{Mass}_m} \tag{D1}$$

with $\text{Mass}_m(x)$ as the mass of particles smaller than $x$ in mode $m$ and $\text{Mass}_m$ as the total particulate mass in mode $m$. $\text{PM}_\text{x}$ is expressed by the modal mass fractions $f_{m,x}$ as follows.

$$\text{PM}_x = \sum_{m \in \{\text{I},\text{J},\text{K}\}} f_{m,x} \times \text{Mass}_m \tag{D2}$$

Because internally mixed particles are assumed in CMAQ, the $\text{PM}_\text{x}$ mass of each individual particulate species can be cal-

15     culated via $f_{m,x}$. As an example, the $\text{PM}_{2.5}$ mass of $\text{NO}_3^-$ is calculated from the modal masses of $\text{NO}_3^-$ given by $P_{m,\text{NO}_3^-}$, $m \in \{\text{I},\text{J},\text{K}\}$:

$$\text{PM}_{2.5,\text{NO}_3^-} = f_{\text{I},2.5} \times P_{\text{I},\text{NO}_3^-} + f_{\text{J},2.5} \times P_{\text{J},\text{NO}_3^-} + f_{\text{K},2.5} \times P_{\text{K},\text{NO}_3^-}$$

The log-normal distributed modes in CMAQ have variable GMDs and $\sigma$s enabling the modes to grow. Because $f_{m,x}$ depends on the particle size distribution, it is also variable. Two different approaches were used to calculate $f_{m,x}$ in this study. The

20     $f_{m,2.5}$ was taken from the CMAQ aerosol diameter diagnostic file. The $f_{m,2.5}$ was calculated following a formula by Jiang et al. (2006). The formula accounts for the loss of particles smaller than 2.5 µm when $\text{PM}_{2.5}$ mass is captured by particle

samplers. In the CMAQ code, the formula is printed as follows.

$$f_{m,2.5} = 0.5 \times \left( 1.0 + \mathrm{erf}\left( \frac{x_{\mathrm{st},m} - \ln \mathrm{GMD}_m}{\sqrt{2} \times \ln \sigma_m} - \frac{3.0 \times \ln \sigma_m}{\sqrt{2}} \right) \right),$$

$$x_{\mathrm{st},m} = 0.5 \times \left( \sqrt{B^2 + 4.0 \times 2.5\,\mu m \times (2.5\,\mu m + B) \times 10^3 \times \rho^{-1}} - B \right)$$

$$B = 0.21470\,\mu m$$

(D3)

where $\mathrm{GMD}_m$ [μm] and $\sigma_m$ (dimensionless) are the geometric mean diameter and standard deviation of the mode $m$, $\rho$ $\left[ kg\,m^{-3} \right]$ is the average particle density, and $x_{\mathrm{st},m}$ [μm] is the Stokes diameter equivalent of the aerodynamic diameter. In CMAQ, the constant $B$ is denoted as the Cunningham slip-correction parameter.

The fractions $f_{m,10}$ and $f_{m,0.1}$ were calculated via integrating the particle mass distribution as shown in Eq. (D4). After canceling some terms and using Eq. (3) of Binkowski and Roselle (2003), one arrives at Eq. (D5), from which $f_{m,x}$ is calculated.

$$f_{m,x} = \frac{\frac{4}{3}\rho \int_0^x D^3 \times \frac{1}{\sqrt{2\pi} \times \ln\sigma_m \times D} \exp\left( -0.5 \left( \frac{\ln D - \ln\mathrm{GMD}_m}{\ln\sigma_m} \right)^2 \right) dD}{\frac{4}{3}\rho \int_0^\infty D^3 \times \frac{1}{\sqrt{2\pi} \times \ln\sigma_m \times D} \exp\left( -0.5 \left( \frac{\ln D - \ln\mathrm{GMD}_m}{2\ln\sigma_m} \right)^2 \right) dD}$$

(D4)

$$= \frac{\int_0^x \frac{D^2}{\sqrt{2\pi} \times \ln\sigma_m} \exp\left( -0.5 \left( \frac{\ln D - \ln\mathrm{GMD}_m}{\ln\sigma_m} \right)^2 \right) dD}{\int_0^\infty \frac{D^2}{\sqrt{2\pi} \times \ln\sigma_m} \exp\left( -0.5 \left( \frac{\ln D - \ln\mathrm{GMD}_m}{\ln\sigma_m} \right)^2 \right) dD}$$

$$= \frac{\int_0^x \frac{D^2}{\sqrt{2\pi} \times \ln\sigma_m} \exp\left( -0.5 \left( \frac{\ln D - \ln\mathrm{GMD}_m}{\ln\sigma_m} \right)^2 \right) dD}{\mathrm{GMD}_m^3 \times \exp\left( 4.5\,(\ln\sigma_m)^2 \right)}$$

$$= \mathrm{GMD}_m^{-3} \times \exp\left( -4.5\,(\ln\sigma_m)^2 \right) \times \int_0^x \frac{D^2}{\sqrt{2\pi} \times \ln\sigma_m} \exp\left( -0.5 \left( \frac{\ln D - \ln\mathrm{GMD}_m}{\ln\sigma_m} \right)^2 \right) dD$$

(D5)

## Appendix E: Extinction Coefficient

The extinction coefficient is calculated according to Eq. (2). The calculation of the individual mass components $P_i$ is presented below.

$$P_{\text{small ammonium sulfate}} = \sum_{m\in\{\text{I,J,K}\}} \left( f_{m,0.1} \times P_{m,\text{SO}_4^{2-}} \right) + \sum_{m\in\{\text{I,J,K}\}} \left( f_{m,0.1} \times r_{m,\text{SO}_4^{2-}} \times P_{m,\text{NH}_4^+} \right) \tag{E1}$$

$$P_{\text{large ammonium sulfate}} = \sum_{m\in\{\text{I,J,K}\}} \left( (f_{m,2.5} - f_{m,0.1}) \times P_{m,\text{SO}_4^{2-}} \right) + \sum_{m\in\{\text{I,J,K}\}} \left( (f_{m,2.5} - f_{m,0.1}) \times r_{m,\text{SO}_4^{2-}} \times P_{m,\text{NH}_4^+} \right) \tag{E2}$$

$$P_{\text{small ammonium nitrate}} = \sum_{m\in\{\text{I,J,K}\}} \left( f_{m,0.1} \times P_{m,\text{NO}_3^-} \right) + \sum_{m\in\{\text{I,J,K}\}} \left( f_{m,0.1} \times r_{m,\text{NO}_3^-} \times P_{m,\text{NH}_4^+} \right) \tag{E3}$$

$$P_{\text{large ammonium nitrate}} = \sum_{m\in\{\text{I,J,K}\}} \left( (f_{m,2.5} - f_{m,0.1}) \times P_{m,\text{NO}_3^-} \right) + \sum_{m\in\{\text{I,J,K}\}} \left( (f_{m,2.5} - f_{m,0.1}) \times r_{m,\text{NO}_3^-} \times P_{m,\text{NH}_4^+} \right) \tag{E4}$$

$$P_{\text{small organic mass}} = \sum_{m\in\{\text{I,J}\}} \left( f_{m,0.1} \times P_{m,\sum\text{org}} \right) \tag{E5}$$

$$P_{\text{large organic mass}} = \sum_{m\in\{\text{I,J}\}} \left( (f_{m,2.5} - f_{m,0.1}) \times P_{m,\sum\text{org}} \right) \tag{E6}$$

$$P_{\text{elemental carbon}} = \sum_{m\in\{\text{I,J}\}} \left( f_{m,2.5} \times P_{m,\text{EC}} \right)$$

$$P_{\text{fine soil}} = f_{\text{J},2.5} \times P_{\text{J,A25}} + f_{\text{K},2.5} \times P_{\text{K,Soil}} + f_{\text{K},2.5} \times P_{\text{K,Cors}} \tag{E7}$$

$$P_{\text{sea salt}} = \sum_{m\in\{\text{I,J,K}\}} \left( f_{m,2.5} \times P_{m,\text{Na}^+} \right) + \sum_{m\in\{\text{I,J,K}\}} \left( f_{m,2.5} \times P_{m,\text{Cl}^-} \right) \tag{E8}$$

$$P_{\text{coarse mass}} = \sum_{m\in\{\text{I,J,K}\}} \left( (f_{m,10} - f_{m,2.5}) \times \left( P_{m,\text{SO}_4^{2-}} + P_{m,\text{NO}_3^-} + P_{m,\text{NH}_4^+} + P_{m,\text{Na}^+} + P_{m,\text{Cl}^-} \right) \right)$$

$$+ \sum_{m\in\{\text{I,J}\}} \left( (f_{m,10} - f_{m,2.5}) \times \left( P_{m,\sum\text{org}} + P_{m,\text{EC}} \right) \right)$$

$$+ (f_{\text{J},10} - f_{\text{J},2.5}) \times P_{\text{J,A25}} + (f_{\text{K},10} - f_{\text{K},2.5}) \times (P_{\text{K,Soil}} + P_{\text{K,Cors}}) \tag{E9}$$

with

$$r_{m,\text{SO}_4^{2-}} = \frac{2 \times P_{m,\text{SO}_4^{2-}} \times \text{M}_{\text{SO}_4^{2-}}}{P_{m,\text{NO}_3^-} \times \text{M}_{\text{NO}_3^-} + 2 \times P_{m,\text{SO}_4^{2-}} \times \text{M}_{\text{SO}_4^{2-}}} \tag{E10}$$

$$r_{m,\text{SO}_4^{2-}} = 1 - r_{m,\text{SO}_4^{2-}} \tag{E11}$$

$$\text{M}_{\text{SO}_4^{2-}} = 96 \text{ g mol}^{-1} \tag{E12}$$

$$\text{M}_{\text{NO}_3^-} = 62 \text{ g mol}^{-1} \tag{E13}$$

$$P_{\text{I},\sum\text{org}} = P_{\text{I,ORGPA}} \tag{E14}$$

$$P_{\text{J},\sum\text{org}} = P_{\text{J,ALK}} + \sum_{i=1}^{3} P_{\text{J,XYL}_i} + \sum_{i=1}^{3} P_{\text{J,TOL}_i} + \sum_{i=1}^{3} P_{\text{J,BNZ}_i} + \sum_{i=1}^{3} P_{\text{J,ISO}_i} + \sum_{i=1}^{2} P_{\text{J,TRP}_i}$$

$$+ P_{\text{J,SQT}} + P_{\text{J,ORGC}} + P_{\text{J,ORGPA}} + P_{\text{J,OLGA}} + P_{\text{J,OLGB}} \tag{E15}$$

$P_{I,\sum org}$ and $P_{J,\sum org}$ summarize all organic compounds, which are included in CMAQ. The abbreviations used in CMAQ are used here to simplify the inspection for CMAQ users. The secondary organic aerosol mechanism of CMAQ and the naming conventions are described in Carlton et al. (2010).

Sulfate is not considered for the calculation of $P_{\text{sea salt}}$. This is not completely correct because sea salt sulfate is *bound* as sodium sulfate and not as ammonium sulfate. However, nitrate that condensed on sea salt particles and displaced chloride is also *bound* as sodium nitrate and not as ammonium nitrate. Hence, it is consistent to put the total sulfate and the total nitrate into ammonium sulfate and ammonium nitrate masses, respectively.

In the $r_{m,SO_4^{2-}}$ calculation, the moles of sulfate ($P_{m,SO_4^{2-}} \times M_{SO_4^{2-}}$) are multiplied by 2 because sulfate as two negative charges (2- = two free electrons), whereas nitrate has only one (- = one free electron). The split of ammonium between ammonium sulfate and ammonium nitrate should be performed on the base of the available negative charges (free electrons).

*Acknowledgements.* For the calculation of sea salt emissions, input data were obtained from the European Centre for Medium-Range Weather Forecasts (ECMWF), the German Federal Maritime and Hydrographic Agency (BSH), and the coastDatII database of Helmholtz-Zentrum Geesthacht (HZG). In particular, Beate Geyer and Nikolaus Groll supported this work by preparing the coastDatII meteorological and wave data, respectively, and by answering several questions about these data. We thank Matthias Karl and Jan Arndt for fruitful discussions and literature references as well as Markus Schultze and Joanna Staneva for providing additional information on the meteorological and wave data. Jim Kelly, Brett Gantt and Uma Shankar (U.S. EPA) answered questions about the implementation of the sea salt emission calculations in CMAQ, and Monica Mårtensson (Uppsala University), Astrid Manders-Groot (TNO) and Jurgita Ovadnevaite (National University of Ireland) provided helpful comments with respect to their sea salt studies and sea salt emission parameterizations. We offer our general gratitude to the U.S. EPA and the CMAQ development team for providing this high-quality chemistry transport model as an open-source product. The EMEP measurement data were extracted from the EBAS database, which is maintained and updated by the Norwegian Institute for Air Research (NILU). In particular, we thank Anne Hjellbrekke for answering questions about the EMEP data. The statistical evaluations and most plotting tasks were performed using R. The remaining plots were created using the Generic Mapping Tools (GMT) developed and maintained by Paul Wessel, Walter H. F. Smith, Remko Scharroo, Joaquim Luis and Florian Wobbe. The simulation data were processed using the Climate Data Operators (CDO) suite developed by Uwe Schultz-Weider from the Max-Planck-Institute for Meteorology and using the netCDF Operators (NCO) suite developed by Charlie Zender and Henry Butowsky. Finally, we thank American Journal Experts (AJE) for English language editing.

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
