# Peer review of "A comparison of sea salt emission parameterizations in Northwestern Europe using a chemistry transport model setup"

_Atmospheric Chemistry and Physics, 2015_

## Referee Comment (RC1) · Anonymous Referee #2 · 24 Mar 2016

Review to the paper "A comparison of sea salt emission parameterizations in Northwestern Europe using a chemistry transport model setup" by Neumann Daniel, Matthias Volker, Bieser Johannes, Aulinger Armin and Quante Markus

In the paper, the comparison of three parameterizations of sea salt emission within the framework of CMAQ chemical transport model is presented. The source functions tested here are those formulated in the works of Gong (2003), Spada et al. (2013) and Ovadnevaite et al. (2014), which calculate sea salt particle production depending on different combinations of wind speed, salinity, sea surface temperature and wave conditions. To evaluate the accuracy of those source functions model calculated sodium concentrations are compared with observations, focusing on North-Western European
regions.

The paper includes ample discussions of the results, thus making a nice contribution to sea salt modelling regarding the choice of parametrizations and demonstrating the uncertainties. Regretfully, the studied period is rather short, being limited to just a few months, January-February and June-August 2008, which makes the conclusions less robust. Moreover, there is significant inter-annual variability in sea salt production related to meteorological variability, and it'd be worth testing the parameterizations for a broader set of conditions. Regarding evaluation of the results for sea salt size distribution, much more measurements are available for later years than 2008, which would facilitate making more founded conclusions. I think that the limited character of the study should be pointed out in relevant places in the paper (abstract, study aim, conclusions). Then, why would not the authors also look at sodium concentration in precipitation, for which much more measurements are available.

Furthermore, I cannot see that the authors manage to achieve the stated aim, namely "to improve modeled atmospheric sea salt concentrations..", but rather "to contribute" to this improvement through testing some of the available source functions, pointing to their strength/weaknesses and recommending certain improvements to the parameterizations (though without priorly testing them) .

Despite the above comments, I consider the paper to be an interesting study and support its publication in ACP after some minor revisions and editing are performed.

1. Does the paper address relevant scientific questions within the scope of ACP? Yes. Accurate modelling of sea salt is important for models' ability to adequately describe atmospheric heterogeneous chemistry and pollutants' residence time, and thus important for assessments of air quality, acidification and eutrophication especially in the coastal regions. The

2. Does the paper present novel concepts, ideas, tools, or data? Partly. The novel piece of work is testing the recently published parameterization by Ovadnevaite et al.

(2014) in comparison with the the other ones commonly used in CTMs and observation.

3. Are substantial conclusions reached? The paper presents findings and discussions regarding comparative performance of the three sea salt parametrizations at different distances from the coasts based on the rather limited observational dataset. Also, some recommendations are made (but not tested in practice) to improve the sea salt source functions.

4. Are the scientific methods and assumptions valid and clearly outlined? Yes

5. Are the results sufficient to support the interpretations and conclusions? Partly. No model experiments were conducted to trial some statements, like hypotheses of discrepancies between the parameterization and observations etc.

6. Is the description of experiments and calculations sufficiently complete and precise to allow their reproduction by fellow scientists (traceability of results)? Yes 7. Do the authors give proper credit to related work and clearly indicate their own new/original contribution? Yes 8. Does the title clearly reflect the contents of the paper? Yes

9. Does the abstract provide a concise and complete summary? Yes. I'd recommend to include in the abstract the time period for which the comparisons were performed.

10. Is the overall presentation well structured and clear? Yes

11. Is the language fluent and precise? Yes, mostly. In several cases, it's better to use "close to each other" or "similar" instead of "similar to each other"

12. Are mathematical formulae, symbols, abbreviations, and units correctly defined and used? Largely Yes. Please, explain SST and SAL line 126.

13. Should any parts of the paper (text, formulae, figures, tables) be clarified, reduced, combined, or eliminated? I'd recommend to a bit shorten section 3.2, perhaps skipping describing all small details and rather offering more summarized findings.

14. Are the number and quality of references appropriate? Yes 15. Is the amount and

quality of supplementary material appropriate? Yes

Other comments:

1. Please, be consistent in using "Na in PMx" instead of just Pmx to avoid misunderstandings.

2. Check the sentence starting p. 1 line 23 (The parameterization of sea salt emissions has a long history..because..because..)

3. P.2 repetition in lines 1-3 and 12-13; lines 15-30 – relevance to PM pollution/ air quality?

4. In sec. 2 (model description), it'd be useful to explain how sea salt dry and wet deposition is calculated in the model. In what heights/model layers the emitted sea salt is distributed? Also, what was the upper cut-off size in the modelled sea salt?

5. Section 2.3: given the strong dependence of sea salt production on wind speed, could the authors say something about the accuracy of the wind data?

6. P. 10 line 5: explain the choice of Spearman's correlation (not Pearson's)

7. P. 12 line 14: change "least highest" to e.g. lowest

8. P. 13 line 1-2: positive/negative bias already means over/under-estimation – no need for repetition

9. P. 15 line 3: similar to; or similar as....

16-22: the obvious stuff; I'd recommend to edit: The analysis REVEALS...

29: at the station Westerland

33: dry deposition rate (instead of behaviour), perhaps

10. P. 17 lines 8-9: this difference of OV14 treatment of surf zone should probably be noted before, while comparing/evaluating the results

line 31: partially over/under-estimate??? maybe better to say "on average"? Or "in general"? 11. P. 18 lines 5-8: it's somewhat unclear to me what the authors are trying to say here

---

## Referee Comment (RC2) · Anonymous Referee #3 · 31 Mar 2016

General Summary:

This article by Neumann and coauthors describes a modeling study in which several sea salt emission parameterizations are evaluated for Northwestern Europe during winter and summer months. While the study does a comprehensive comparison of the different parmeterizations, evaluation of the parameterizations with observational data is rather limited. As this region has recently been shown by the same authors to have an important interaction between sea salt and nitrogen deposition, evaluating and improving the sea salt emissions in the model is important. I'd recommend publication in ACP after the following comments listed below have been addressed.

Major comments:

1) I consider the Sect 3.2.2 the weakest part of the article due to the limited observational dataset of particle size distributions. Without size resolved observations besides coincident PM2.5 and PM10 sodium concentrations at one site, little evaluation of the particle size distribution predicted by these emission parameterizations is possible. I'd suggest limiting discussion in Section 3.2.2 to the Melpitz site.

2) I am concerned about comparison of model predictions and PM10 observations throughout the article. Depending on the coarse mode GMD and standard deviation, it seems possible that a significant fraction of the mass could be in particles greater than 10 micrometers in diameter. In addition to describing the environmental conditions of the PM10 measurements (i.e. ambient vs standard temperature and pressure), I'd appreciate the authors analyzing the fraction of the predicted coarse mode that may not have been measured at the various sites. If significant, the authors should consider providing additional statistics for the model-predicted PM10 sodium.

3) In light of the limited observational dataset beyond sodium PM10 concentrations, I'd suggest the authors explore additional datasets that can be used for model evaluation such as satellite-derived aerosol optical depth. Given the large mass and number emission differences between the parameterizations, it may be possible to evaluate whether the associated AOD changes bring the model into better agreement with the satellite-derived values particularly over the Atlantic Ocean.

Minor comments:

1) Page 1, Line 15: should be "coastal"

2) Page 1, Lines 16-18: I don't think the accuracy of fine and coarse mode predictions can be adequately determined by the available dataset and I'd suggest removing or changing the sentence to discuss the comparison between parameterizations.

3) Page 2, Line 25: should be NOx and SO2 where the x and 2 are subscripts

4) Page 3, Line 2: change to something like "This parameterization has not been used

in a CTM setup in the study region."

5) Page 3, Lines 4-9: please expand the discussion of previous sea salt studies in this region

6) Page 3, Figure 1: consider removing the blue and yellow colors denoting the ocean and land

7) Page 3, Sect. 2.1: please state whether dust emissions were active in the model and whether they have an impact on sodium concentrations

8) Page 7, Figure 3: consider using the same x- and y-axes as Figure 2 and give the wind speed used to determine the size distribution

9) Page 9, Figure 4: please fix the label for the Reynolds number that is cut-off

10) Page 9, Figure 5: please explain here or in the text why these three stations were selected as representative stations

11) Page 30, Table 3: consider grouping the stations by coastal and inland sites

---

## Author Comment (AC1) · 1 Jul 2016

**Response to the comments of Reviewer #2**

We thank the reviewer for the constructive comments on the manuscript.

**Major comments**

0. The paper includes ample discussions of the results, thus making a nice contribution to sea salt modelling regarding the choice of parametrizations and demonstrating the uncertainties. Regretfully, the studied period is rather short,

being limited to just a few months, January-February and June-August 2008, which makes the conclusions less robust. Moreover, there is significant inter-annual variability in sea salt production related to meteorological variability, and it'd be worth testing the parameterizations for a broader set of conditions. Regarding evaluation of the results for sea salt size distribution, much more measurements are available for later years than 2008, which would facilitate making more founded conclusions. I think that the limited character of the study should be pointed out in relevant places in the paper (abstract, study aim, conclusions). Then, why would not the authors also look at sodium concentration in precipitation, for which much more measurements are available.

Furthermore, I cannot see that the authors manage to achieve the stated aim, namely "to improve modeled atmospheric sea salt concentrations..", but rather "to contribute" to this improvement through testing some of the available source functions, pointing to their strength/weaknesses and recommending certain improvements to the parameterizations (though without priorly testing them).

> The sentences "Each two months in winter and summer 2008 were considered for this purpose. The shortness of these periods limits generalizability of the conclusions on other years." were added in the abstract. A final paragraph mentioning this topic and giving an outlook on "What should be done next" was added to the conclusions.

> A comparison of sodium concentrations in precipitation was added (Sect. 3.3).

> Statements such as "to improve modeled atmospheric sea salt concentrations.." were altered.

**9. Does the abstract provide a concise and complete summary? Yes. I'd recommend to include in the abstract the time period for which the comparisons were performed.**

> Included in the end of the second paragraph of the Abstract.

**11. Is the language fluent and precise? Yes, mostly. In several cases, it's better to use "close to each other" or "similar" instead of "similar to each other"**

> Replaced as suggested.

**12. Are mathematical formulae, symbols, abbreviations, and units correctly defined and used? Largely Yes. Please, explain SST and SAL line 126.**

> It was not absolutely clear which location in the text was indicated by line 126. An explanation of SAL and SST was added in line 6 on page 6 (which is approximately line 126) and an explanation of SAL to the caption of Table 1.

**13. Should any parts of the paper (text, formulae, figures, tables) be clarified, reduced, combined, or eliminated? I'd recommend to a bit shorten section 3.2, perhaps skipping describing all small details and rather offering more summarized findings.**

> Section 3.2 is shorted (as requested by Reviewer #3, too).

**Other comments**

**1. Please, be consistent in using "Na in PMx" instead of just PMx to avoid misunder- standings.**

> Suggestion included. "sodium" was added at most locations and "$Na^+$" only at a few, because the usage of the first seemed to yield a better reading flow.

**2. Check the sentence starting p. 1 line 23 (The parameterization of sea salt emissions has a long history..because..because..)**

> The sentence was split into two sentences and slightly modified. It now reads: "*The parameterization of sea salt emissions has a long history ((e.g., Blanchard and Woodcock, 1980; Fairall et al., 1983; Monahan and Muircheartaigh, 1980))*. **Such** *parameterizations are necessary in chemistry transport models (CTMs) and climate models* **because sea salt particles** *impact on atmospheric processes.*"

**3. P.2 repetition in lines 1-3 and 12–13; lines 15–30 – relevance to PM pollution/ air quality?**

> The lines 1–3 and 12–13 were merged at the location of lines 12–13 and a new reference (Chen et al., 2016) was added.

> Lines 15–23 were removed, line 26 slightly extended, and references from above were included in line 27. This passage (l.24–28) is very relevant because it gives a major reason why improving sea salt emissions in CTMs is important (except for just predicting sea salt concentrations with higher accuracy for the reason itself).

**4. In sec. 2 (model description), it'd be useful to explain how sea salt dry and wet deposition is calculated in the model. In what heights/model layers the emitted sea salt is distributed? Also, what was the upper cut-off size in the modelled sea salt?**

> Information on the dry deposition parameterization (Binkowski and Shankar, 1995) was added in the end of the first paragraph of Sect. 2.1 (Chemistry Transport Model). Sea salt is emitted into the bottom layer of the model grid (not in higher layers). This

information was added in the end of the first paragraph of Sect. 2.2.3 (Technical Implementation). The sea salt particle emissions as well as the particle concentrations are represented by log-normal distributions with variable geometric mean diameters and standard deviations. No hard-coded upper cut of size exists.

**5. Section 2.3: given the strong dependence of sea salt production on wind speed, could the authors say something about the accuracy of the wind data?**

> We assume that the wind speed is well reproduced and neither strongly over estimated nor strongly underestimated. The following sentence was added to the manuscript: "*The* $10$ m *wind speed is well reproduced above the North Sea (Geyer, 2014; Geyer et al., 2015)*.".

**6. P. 10 line 5: explain the choice of Spearman's correlation (not Pearson's)**

> It should be Pearson's correlation coefficient. It was changed in the text.

**7. P. 12 line 14: change "least highest" to e.g. lowest**

> modified

**8. P. 13 line 1-2: positive/negative bias already means over/under-estimation – no need for repetition**

> The second sentence was removed.

**9.a) P. 15 line 3: similar to; or similar as....**

**9.b) 16-22: the obvious stuff; I'd recommend to edit: The analysis REVEALS...**

**9.c) 29: at the station Westerland**

**9.d) 33: dry deposition rate (instead of behaviour), perhaps**

> a) changed "*similar data than*" to "*similar data as*"

> b) inserted an "as": "*on a similar level **as** at the station Melpitz*"; removed duplications; removed "this analysis reveals" completely;

> c) corrected

> d) corrected

**10.a) P. 17 lines 8-9: this difference of OV14 treatment of surf zone should probably be noted before, while comparing/evaluating the results**

**10.b) line 31: partially over/under-estimate??? maybe better to say "on average"? Or "in general"?**

> a) A paragraph on the surf zone emission treatment starting with "Another important aspect in sea salt modeling studies is the consideration of surf zone emissions . . ." was added in Sect. 3.2.1.

> b) replaced "partially" by "in general"

**11. P. 18 lines 5-8: it's somewhat unclear to me what the authors are trying to say here**

> We removed the two sentences.

---

## Author Comment (AC2) · 1 Jul 2016

**Response to the comments of Reviewer #3**

We thank the reviewer for the constructive comments on the manuscript.

**Major comments**

**1) I consider the Sect 3.2.2 the weakest part of the article due to the limited observational dataset of particle size distributions. Without size resolved observations besides coincident PM2.5 and PM10 sodium concentrations at one site,**

[Figure]

**little evaluation of the particle size distribution predicted by these emission pa-
rameterizations is possible. I'd suggest limiting discussion in Section 3.2.2 to
the Melpitz site.**

> Section 3.2.1 was shortened. The parts on the Waldhof station and on the geometric
mean diameters were removed. The figures were moved into the Supplement. The
part on model results at Westerland was shorted but kept because it highlights how
the sea salt particle size distribution changes from the coast to the hinterland (at least
in the model).

**2) I am concerned about comparison of model predictions and $PM_{10}$ observa-
tions throughout the article. Depending on the coarse mode GMD and standard
deviation, it seems possible that a significant fraction of the mass could be in
particles greater than 10 micrometers in diameter. In addition to describing the
environmental conditions of the PM10 measurements (i.e. ambient vs standard
temperature and pressure), I'd appreciate the authors analyzing the fraction of
the predicted coarse mode that may not have been measured at the various sites.
If significant, the authors should consider providing additional statistics for the
model-predicted PM10 sodium.**

> Instead of comparing the total model particulate sodium mass with sodium $PM_{10}$
measurements as done in the reviewed version, the evaluation was repeated with mod-
eled sodium $PM_{10}$ concentrations. The modeled $PM_{10}$ concentrations were calculated
on the base of the modal distributions (see new Appendix D). As a result, the overesti-
mations in the SP13 and GO03 cases were considerably reduced.

> Unfortunately, no measurement data on ambient temperature and pressure were
available via the EMEP database. Therefore, no evaluation of these parameters was
possible.

**3) In light of the limited observational dataset beyond sodium** $PM_{10}$ **concentrations, I'd suggest the authors explore additional datasets that can be used for model evaluation such as satellite-derived aerosol optical depth. Given the large mass and number emission differences between the parameterizations, it may be possible to evaluate whether the associated AOD changes bring the model into better agreement with the satellite-derived values particularly over the Atlantic Ocean.**

> A section on the comparison with AERONET aerosol optical depth (AOD) measurements was added (Sect. 3.4). We decided against a comparison with satellite-derived AOD data because we are inexperienced in the usage of satellite products and pitfalls with respect to these. > Additionally, we see no added value in adding a comparison with satellite based AOD data because the deviation between the AODs of the model cases is considerably lower than the differences between model and measured AODs. One can expect similar results for spatially resolved AOD data.

**Minor comments:**

**1) Page 1, Line 15: should be "coastal"**

> corrected

**2) Page 1, Lines 16-18: I don't think the accuracy of fine and coarse mode predictions can be adequately determined by the available dataset and I'd suggest removing or changing the sentence to discuss the comparison between parameterizations.**

> removed

[Figure]

**3) Page 2, Line 25: should be NOx and SO2 where the x and 2 are subscripts**

> corrected

**4) Page 3, Line 2: change to something like "This parameterization has not been used in a CTM setup in the study region."**

> modified accordingly

**5) Page 3, Lines 4-9: please expand the discussion of previous sea salt studies in this region**

> A comparison of the findings in this study with the findings in some of the listed studies was added in the end of the Sect. 3.2.1 and in the end of the new Sect. 3.3. Presenting these studies' results in the Introduction would have considerably extended the Introduction section.

> Results of a recently in ACPD published study (Chen et al., 2016) were included.

**6) Page 3, Figure 1: consider removing the blue and yellow colors denoting the ocean and land**

> removed

**7) Page 3, Sect. 2.1: please state whether dust emissions were active in the model and whether they have an impact on sodium concentrations**

> Dust is not considered. A paragraph was added: "*Dust was neither included in the boundary conditions nor in the emissions. The dust concentrations in Northwestern Europe are low compared to sea salt and anthropogenic particle concentrations (Cuevas et al., 2015). Moreover, in episodes with high dust loading, Sahara dust is commonly transported in higher atmospheric layers across Northwestern Europe (MACC II data, obtained from http:// macc.copernicus-atmosphere.eu).*".

**8) Page 7, Figure 3: consider using the same x- and y-axes as Figure 2 and give the wind speed used to determine the size distribution**

> modified accordingly

**9) Page 9, Figure 4: please fix the label for the Reynolds number that is cut-off**

> The label is fixed.

**10) Page 9, Figure 5: please explain here or in the text why these three stations were selected as representative stations**

> A sentence was added: "*The station of Westerland is located directly at the North Sea coast, the station of Zingst is located at the Baltic Sea coast, and the station of Waldhof is located approximately $200\,\mathrm{km}$ inland. Hence, these stations' measurements cover three different sea salt emission regimes.*"

**11) Page 30, Table 3: consider grouping the stations by coastal and inland sites**

> grouped as requested (in all tables)
* * *

---

## Author Response (AR1)

[revised manuscript text omitted]

$$
\frac{dF_{\mathrm{MA03}}}{dD_{\mathrm{dry}}} = W \times (A \times \mathrm{SST} + B)
\tag{B4}
$$

$$
\begin{aligned}
A &= c_4 \times D_{\mathrm{dry}}^4 + c_3 \times D_{\mathrm{dry}}^3 + c_2 \times D_{\mathrm{dry}}^2 + c_1 \times D_{\mathrm{dry}} + c_0 \\
B &= d_4 \times D_{\mathrm{dry}}^4 + d_3 \times D_{\mathrm{dry}}^3 + d_2 \times D_{\mathrm{dry}}^2 + d_1 \times D_{\mathrm{dry}} + d_0
\end{aligned}
$$

The parameterization is valid on the size range $0.02\ \mu m \leq r_{80} \leq 2.8\ \mu m$.

$$
\frac{dF_{\mathrm{SP13}}}{dD_{\mathrm{dry}}} =
\begin{cases}
\frac{dF_{\mathrm{MA03}}}{dD_{\mathrm{dry}}} & D_{\mathrm{dry}} \leq 2.8\ \mu m \\
\frac{dF_{\mathrm{MO86}}}{dD_{\mathrm{dry}}} & D_{\mathrm{dry}} > 2.8\ \mu m \\
& \wedge u_{10} < 9\ \mathrm{m\ s^{-1}} \\
\max\left(\frac{dF_{\mathrm{MO86}}}{dD_{\mathrm{dry}}}, \frac{dF_{\mathrm{SM93}}}{dD_{\mathrm{dry}}}\right) & D_{\mathrm{dry}} > 2.8\ \mu m \\
& \wedge u_{10} \geq 9\ \mathrm{m\ s^{-1}}
\end{cases}
\tag{B5}
$$

SP13 is valid on the size range $0.02\ \mu m \leq D_{\mathrm{dry}} \leq 30\ \mu m$. The parameters $c_i$ and $d_i$ are provided in Table S1.

**B3 OV14**

The sea salt  emission parameterization OV14  reported by Ovadnevaite et al. (2014) is given by Eq. (B6).

$$\frac{dF_{\text{OV14}}}{d\log_{10} D_{\text{dry}}} = \sum_{i=1}^{5} \frac{F_i(\text{Re}_{\text{Hw}})}{\sqrt{2\pi} \times \log_{10}\sigma_i} \times \exp\left(-\frac{1}{2}\left(\frac{\log_{10}\frac{D_{\text{dry}}}{\text{GMD}_i}}{\log_{10}\sigma_i}\right)\right) \tag{B6}$$

$$\text{Re}_{\text{Hw}} = \frac{u_* \times H_S}{\nu_W} = \frac{\sqrt{C_D} \times u_{10} \times H_S}{\nu_W} \tag{B7}$$

The kinetic viscosity $\nu_W$ is calculated according to Eqs. (22) and (8) in Sharqawy et al. (2010). The source function is valid on the size range $0.015\,\mu\text{m} < D_{\text{dry}} < 6\,\mu\text{m}$. The values for $\text{GMD}_i$, $\sigma_i$, and $F_i$ are given in the supplement (Table S2) and in Ovadnevaite et al. (2014).

**Appendix C: Statistical Evaluation**

The statistical figures residual absolute error (RAE), mean normalized bias (MNB), and Pearson's correlation coefficient (R) are calculated according to Eqs.  (C1), (C2), and (C3), respectively.

$$\text{RAE} = \frac{1}{n} \times \sum_{i=1}^{n} |P_i - O_i| \tag{C1}$$

$$\text{MNB} = \frac{1}{n} \times \sum_{i=1}^{n} \frac{P_i - O_i}{O_i} \tag{C2}$$

$$\text{R} = 1 - \frac{6}{n(n^2-1)} \times \sum_{i=1}^{n} (P_i - O_i)^2 \tag{C3}$$

with

| | |
|---|---|
| $P_i$ | i[th] predicted value |
| $p_i$ | rank of the i[th] predicted value |
| $O_i$ | i[th] observed value |
| $o_i$ | rank of the i[th] observed value |
| $n$ | number of observations |

**Appendix D: Calculating $\text{PM}_{\text{x}}$**

In CMAQ, the particle mass is represented by three log-normal distributed modes denoted as I, J, and K modes – Aitken, accumulation, and coarse mode, respectively. The $\text{PM}_{\text{x}}$ masses of the individual modes need to be obtained and summed to

obtain the $PM_x$ mass over all modes. The latter can be calculated by the modal mass fractions $f_{m,x}$:

$$f_{m,x} = \frac{\text{Mass}_m(x)}{\text{Mass}_m} \tag{D1}$$

with $\text{Mass}_m(x)$ as the mass of particles smaller than $x$ in mode $m$ and $\text{Mass}_m$ as the total particulate mass in mode $m$. $PM_x$ is expressed by the modal mass fractions $f_{m,x}$ as follows.

$$PM_x = \sum_{m \in \{I,J,K\}} f_{m,x} \times \text{Mass}_m \tag{D2}$$

Because internally mixed particles are assumed in CMAQ, the $PM_x$ mass of each individual particulate species can be calculated via $f_{m,x}$. As an example, the $PM_{2.5}$ mass of $NO_3^-$ is calculated from the modal masses of $NO_3^-$ given by $P_{m,NO_3^-}, m \in \{I,J,K\}$:

$$PM_{2.5,NO_3^-} = f_{I,2.5} \times P_{I,NO_3^-} + f_{J,2.5} \times P_{J,NO_3^-} + f_{K,2.5} \times P_{K,NO_3^-}$$

10 The log-normal distributed modes in CMAQ have variable GMDs and $\sigma$s enabling the modes to grow. Because $f_{m,x}$ depends on the particle size distribution, it is also variable. Two different approaches were used to calculate $f_{m,x}$ in this study. The $f_{m,2.5}$ was taken from the CMAQ aerosol diameter diagnostic file. The $f_{m,2.5}$ was calculated following a formula by Jiang et al. (2006). The formula accounts for the loss of particles smaller than $2.5\,\mu m$ when $PM_{2.5}$ mass is captured by particle samplers. In the CMAQ code, the formula is printed as follows.

$$f_{m,2.5} = 0.5 \times \left(1.0 + \text{erf}\left(\frac{x_{st,m} - \ln \text{GMD}_m}{\sqrt{2} \times \ln \sigma_m} - \frac{3.0 \times \ln \sigma_m}{\sqrt{2}}\right)\right), \tag{D3}$$

$$x_{st,m} = 0.5 \times \left(\sqrt{B^2 + 4.0 \times 2.5\,\mu m \times (2.5\,\mu m + B) \times 10^3 \times \rho^{-1}} - B\right)$$

$$B = 0.21470\,\mu m$$

where $\text{GMD}_m$ [$\mu m$] and $\sigma_m$ (dimensionless) are the geometric mean diameter and standard deviation of the mode $m$, $\rho$ [$kg\,m^{-3}$] is the average particle density, and $x_{st,m}$ [$\mu m$] is the Stokes diameter equivalent of the aerodynamic diameter. In

20 CMAQ, the constant $B$ is denoted as the Cunningham slip-correction parameter.

The fractions $f_{m,10}$ and $f_{m,0.1}$ were calculated via integrating the particle mass distribution as shown in Eq. (D4). After canceling some terms and using Eq. (3) of Binkowski and Roselle (2003), one arrives at Eq. (D5), from which $f_{m,x}$ is calculated.

$$f_{m,x} = \frac{\frac{4}{3}\rho \int_0^x D^3 \times \frac{1}{\sqrt{2\pi} \times \ln\sigma_m \times D} \exp\left(-0.5\left(\frac{\ln D - \ln \text{GMD}_m}{\ln\sigma_m}\right)^2\right) dD}{\frac{4}{3}\rho \int_0^\infty D^3 \times \frac{1}{\sqrt{2\pi} \times \ln\sigma_m \times D} \exp\left(-0.5\left(\frac{\ln D - \ln \text{GMD}_m}{2\ln\sigma_m}\right)^2\right) dD}$$

(D4)

$$= \frac{\int_0^x \frac{D^2}{\sqrt{2\pi} \times \ln\sigma_m} \exp\left(-0.5\left(\frac{\ln D - \ln \text{GMD}_m}{\ln\sigma_m}\right)^2\right) dD}{\int_0^\infty \frac{D^2}{\sqrt{2\pi} \times \ln\sigma_m} \exp\left(-0.5\left(\frac{\ln D - \ln \text{GMD}_m}{\ln\sigma_m}\right)^2\right) dD}$$

$$= \frac{\int_0^x \frac{D^2}{\sqrt{2\pi} \times \ln\sigma_m} \exp\left(-0.5\left(\frac{\ln D - \ln \text{GMD}_m}{\ln\sigma_m}\right)^2\right) dD}{\text{GMD}_m^3 \times \exp\left(4.5\left(\ln\sigma_m\right)^2\right)}$$

[revised manuscript text omitted]